# Pericytes are progenitors for coronary artery smooth muscle

**Katharina S Volz[1,2,3], Andrew H Jacobs[2], Heidi I Chen[2], Aruna Poduri[2], Andrew S McKay[2], Daniel P Riordan[4], Natalie Kofler[5], Jan Kitajewski[5], Irving Weissman[3,6], Kristy Red-Horse[2]\***

[1]Stem Cell and Regenerative Medicine PhD Program, Stanford School of Medicine, Stanford, United States; [2]Department of Biological Sciences, Stanford University, Stanford, United States; [3]Institute for Stem Cell and Regenerative Medicine, Stanford School of Medicine, Ludwig Center, Stanford, United States; [4]Department of Biochemistry, Stanford School of Medicine, Stanford, United States; [5]Columbia University Medical Center, New York, United States; [6]Ludwig Center for Cancer Stem Cell Biology and Medicine at Stanford University, Stanford, United States

**Abstract** Epicardial cells on the heart's surface give rise to coronary artery smooth muscle cells (caSMCs) located deep in the myocardium. However, the differentiation steps between epicardial cells and caSMCs are unknown as are the final maturation signals at coronary arteries. Here, we use clonal analysis and lineage tracing to show that caSMCs derive from pericytes, mural cells associated with microvessels, and that these cells are present in adults. During development following the onset of blood flow, pericytes at arterial remodeling sites upregulate Notch3 while endothelial cells express Jagged-1. Deletion of Notch3 disrupts caSMC differentiation. Our data support a model wherein epicardial-derived pericytes populate the entire coronary microvasculature, but differentiate into caSMCs at arterial remodeling zones in response to Notch signaling. Our data are the first demonstration that pericytes are progenitors for smooth muscle, and their presence in adult hearts reveals a new potential cell type for targeting during cardiovascular disease.

**\*For correspondence:** kredhors@stanford.edu

**Competing interests:** The authors declare that no competing interests exist.

## Introduction

Coronary artery disease is the leading cause of death worldwide, but there is currently no effective method to regenerate new coronary arteries (CA) in diseased or injured hearts (*Rubanyi, 2013*). This is likely due to our limited understanding of CA progenitor cells and the signaling pathways that activate their differentiation. CAs are the vessels that supply blood to ventricular heart muscle and are composed of an inner endothelial cell lining wrapped by a smooth muscle covering. Coronary artery smooth muscle cells (caSMC) are particularly important due to their role in the pathogenesis of coronary artery disease. However, caSMC development, both normally during embryogenesis and ectopically during coronary artery disease, is a poorly understood process.

CaSMC development occurs during arterialization of immature coronary plexus vessels. In the murine heart, coronary endothelial cells derived primarily from the sinus venosus and endocardial cells that sprout onto the heart to form an early vascular plexus (*Kattan et al., 2004*; *Chen et al., 2014b*; *Red-Horse et al., 2010*; *Tian et al., 2013*; *Wu et al., 2012*). A subset of the plexus vessels differentiates into arteries once the network attaches to the aorta and begins receiving blood flow (*Chen et al, 2014a*; *Hood et al., 1992*; *Peeters et al., 1997*). Although ultimately adjacent to CA endothelial cells, the source of caSMCs is different. These cells arise from the mesothelial covering of the heart called the epicardium (*Cai et al., 2008b*; *Mikawa and Gourdie, 1996*; *Wilm et al.,*

**eLife digest** The heart is a complex organ composed of several different cell types. Muscle cells of walls of the heart contract to pump blood around the body. These muscle cells are themselves supplied with blood from the coronary arteries that penetrate deep into this muscle tissue. The lining of the coronary arteries is made of endothelial cells, while smooth muscle cells (or SMCs for short) surround the arteries and provide support. The SMCs can also contract to increase or decrease blood flow to the heart, depending on the heart rate.

Endothelial cells and SMCs of the coronary arteries physically interact but develop from different precursor cells. The coronary artery SMCs are derived from cells that comprise the outer layer of the heart (called the epicardium) and move inwards during embryonic development. However, it was not clear exactly what kind of cells these precursor cells are, or which molecular signals trigger their conversion into SMCs.

Volz et al. have studied cardiac development in mice and used fluorescent labels to observed individual cells of the epicardium as they divided and moved. This revealed that when epicardial cells developed into the coronary artery SMCs, there was always an intermediate cell type that wrapped around the developing blood vessels. Upon further investigation, Volz et al. found that these cells were so-called pericytes, which otherwise support small blood vessels throughout the body. Furthermore, the pericytes that did not develop into SMCs remained near the coronary arteries and were still present in adult hearts. Lastly, experiments showed that a protein called Notch-3 is expressed on pericytes and interacts with another protein called Jagged-1 on endothelial cells to prompt the conversion of pericytes into SMCs.

Since heart development is similar in mice and humans, these findings may have implications for future therapies of coronary artery disease, the most common cause of death worldwide. Currently there are no methods to trigger the formation of new coronary arteries after injury or blockage, but knowledge of the pericyte precursors and the signaling pathways that turn them into SMCs could eventually lead to new treatments.

*2005*; *Zhou et al., 2008*). In rodents, caSMCs reside deep within the myocardium while epicardial cells are located on the outermost layer of the heart. During embryonic development, many epicardial cells undergo an epithelial-to-mesenchymal transition (EMT) and migrate into the deeper layers of the myocardium to form the stromal cells of the heart, including cardiac fibroblasts and caSMCs. Inhibition of epicardial migration from the surface by deletion of Platelet Derived Growth Factor Receptor β (PDGFRβ) diminishes caSMC development (*Mellgren, et al., 2008*; *Smith et al., 2011*). However, the migrating epicardial-derived cell type fated to become caSMCs has not been discovered, and the mechanisms that trigger this intermediate progenitor to differentiate into caSMCs are unknown.

Smooth muscle within other internal organs, including vascular smooth muscle, also arises from an outer mesothelial covering. Lineage tracing Mesothelin (Msln)-positive mesothelial cells and their prospective isolation and transplantation has shown that many of the abdominal and thoracic organs derive their smooth muscle and fibroblasts from the surface serosa layer (*Rinkevich et al., 2012*).. Mesothelial to vascular smooth muscle differentiation occurred almost exclusively during developmental and early postnatal stages, but, similar to the heart, the cellular pathway bridging the surface to internal arteries has not been identified. Epicardial mesothelium of the heart also has a developmental restriction. Most migrating caSMC progenitors leave the heart surface before embryonic day 12.5 and are no longer able to migrate into the adult heart, either normally or following myocardial infarction (*Wei et al., 2015*; *Zhou et al., 2011*). However, since this time point is long before caSMCs appear, it is unclear what cell type resides in the heart to eventually receive signals to become new smooth muscle around forming arteries and whether this cell type persists in the adult. The discovery of such an intermediate progenitor could identify a cell type that could aid collateral artery formation during disease.

Here, we find that cells resembling vascular pericytes (PDGFRβ$^+$Notch3$^+$NG2$^+$SM-MHC$^-$SMα$^-$PDGFRα$^-$, wrapping microvessels, embedded within a basement membrane) are intermediate progenitors for smooth muscle in the developing heart. Pericytes are mural cells that wrap

microvascular blood vessels and regulate their development and function (*Armulik et al., 2011*; *Cappellari et al., 2013*). Pericytes share features with smooth muscle cells including close apposition to the vessel and some molecular markers, but differ in their contractile protein expression, cell shape, location on capillaries instead of large vessels, and discontinuous covering of the endothelium. Since their identification in 1873 (*Rouget, 1873*), biologists have wondered whether pericytes and smooth muscle differentiate into each other, but direct evidence has been lacking (*Armulik et al., 2011*; *Cappellari et al., 2013*; *Majesky, 2011*). We provide evidence that pericytes lining the coronary vascular plexus respond to Notch3 signaling at arterial remodeling zones to become mature caSMCs during embryonic development. We also observed caSMC related pericytes in the adult heart presenting the possibility that the Notch3 pathway could be manipulated in these cells to participate in CA regeneration.

## Results

### CaSMC development

To identify where an epicardial-derived caSMC progenitor would be found, we determined where caSMCs first appear. CA development was followed using confocal imaging of intact mouse hearts immunostained for vascular endothelial-cadherin (VE-cadherin) and smooth muscle-myosin heavy chain (SM-MHC), one of the most specific markers for mature smooth muscle (*Miano et al., 1994*; *Seidelmann et al., 2013*). Hearts are shown on the right lateral side to most effectively display developing CAs. Development of CAs begins with the invasion of VE-cadherin$^+$ endothelial cells that form an immature coronary vascular plexus (*Kattan et al., 2004*; *Chen et al., 2014b*; *Red-Horse et al., 2010*), which is initially devoid of blood flow (*Figure 1A–A'''*) (*Chen et al., 2014a*). At embryonic day (e) 13.5, the plexus vessels attach to the aorta and begin to receive blood flow (*Chen et al., 2014a*). Subsequently, vascular remodeling, or fusion and enlargement of plexus vessels, is observed directly downstream of the aortic attachment site where future arteries will form (*Chen et al., 2014a*). SM-MHC protein expression was first observed at e14.5 in plexus vessels that had begun to remodel into arteries (*Figure 1B–B'''*). At e14.5 smooth muscle coverage was patchy and consisted of cells with both low and high SM-MHC (SM-MHC$^{low}$ or SM-MHC$^{high}$) protein expression (fluorescence measured from single confocal z-planes) (*Figure 1B'''*, *Figure 1—figure supplement 1*). As this initial remodeling zone transitioned into recognizable arterial vessels at e15.5, caSMC coverage increased and most cells were SM-MHC$^{high}$ (*Figure 1C–C'''*). Remodeling zones with patchy SM-MHC$^{low}$ cells were now just distal to the more mature vessels (*Figure 1C'''*). These distal remodeling zones eventually transformed into mature vessels in the next developmental stage at e16.5 (*Figure 1D–D'''*). These data identify when and where SM-MHC$^+$ caSMCs differentiate within the context of the whole heart at single-cell resolution (*Figure 1E*).

The temporal appearance and location, i.e. at the remodeling zone early in arterial development, of SM-MHC$^{low}$ cells suggested that they have recently initiated caSMC differentiation while cells expressing higher SM-MHC levels were more mature (*Figure 1F,G*). Measuring the distance between SM-MHC$^{low}$ cells and the epicardium revealed that they were not on the surface of the heart, but on average 50 µm deep (*Figure 1H*). Thus, a putative intermediate progenitor would need to migrate through approximately 7 cell layers between the epicardium and nascent CAs where they first express the mature smooth muscle cell marker (*Figure 1I*).

### Clonal analysis of epicardial-derived cells

Because most caSMCs derive from the epicardium (*Cai et al., 2008b*; *Mikawa and Gourdie, 1996*; *Pérez-Pomares et al., 2002*; *Wilm et al., 2005*; *Zhou et al., 2008*), we aimed to find the epicardial-derived progenitor that eventually differentiates into caSMCs. Epicardial-driven Cre-expressing mice for lineage tracing exist; however, these constructs label all the other epicardial-derived stromal cells in the heart in addition to caSMCs. We needed to study the smooth muscle lineage in isolation, which is done most accurately using clonal level labeling where single cells and all their progeny are genetically marked with a fluorescent tag (*Buckingham and Meilhac, 2011*). Our approach was to identify caSMC progenitors by analyzing fluorescently labeled sister cells within clonal clusters that also contain caSMCs. To produce clones, epicardial cells were sparsely labeled using T-box 18 (Tbx18)-Cre (*Cai et al., 2008b*) coupled with the Mosaic Analysis with Double Markers (MADM) Cre

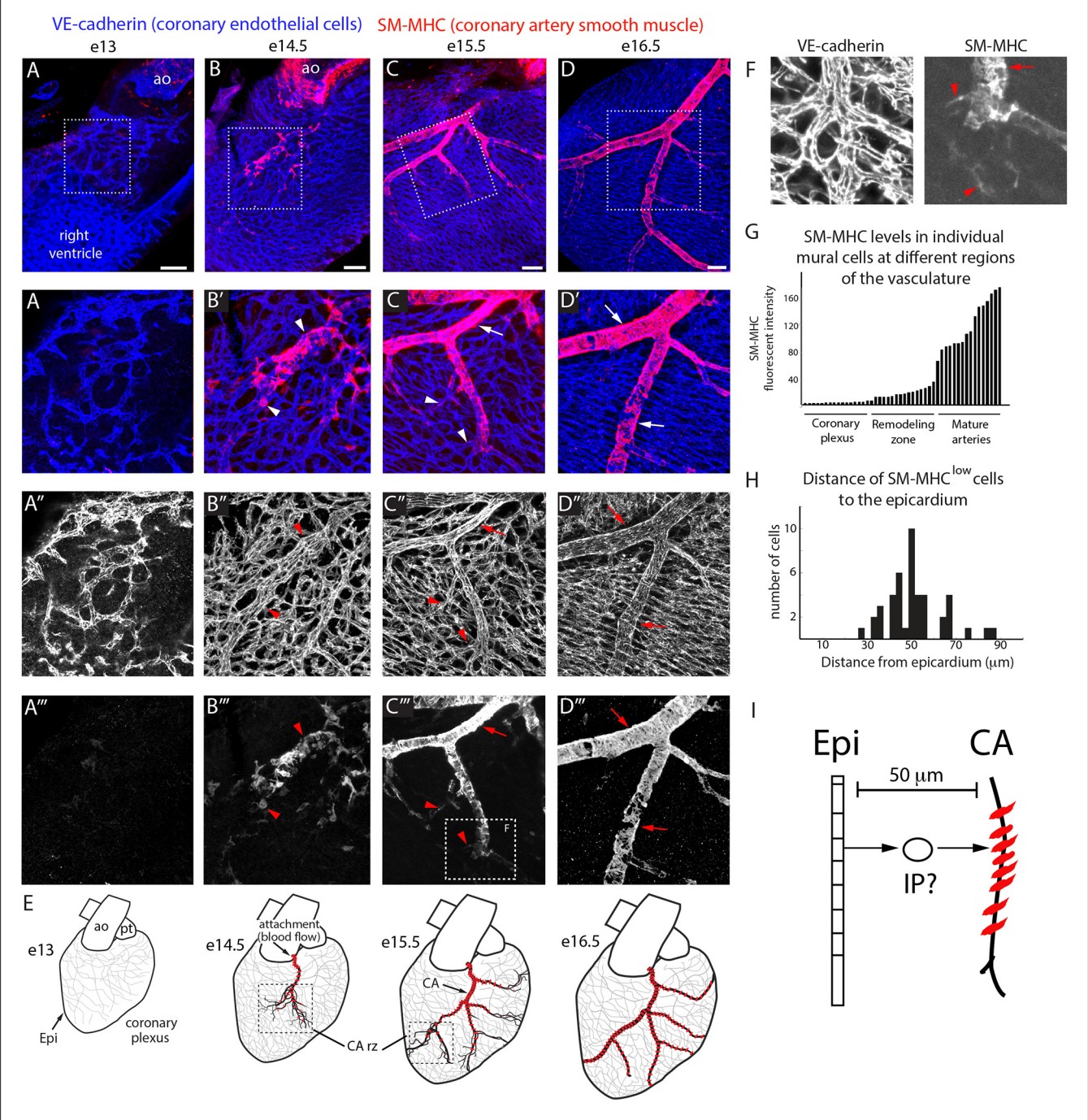

**Figure 1.** Coronary artery smooth muscle differentiation is initiated early during vascular remodeling. (A–D) Whole mount confocal images of the developing right coronary artery at embryonic (e) days 13 (A–A'''), 14.5 (B–B'''), 15.5 (C–C'''), and 16.5 (D–D''') immunostained with VE-cadherin (blue) and SM-MHC (red). Higher magnification views (z-stack subsets) from boxed regions (A'–D') and separated channels (A''–D'') show that smooth muscle first appears at early remodeling zones (arrowheads) and further accumulates as these transform into coronary arteries (arrows). (E) Schematic representation showing coronary artery (CA) smooth muscle cell (red) development coincident with aortic attachment and initiation of blood flow at the coronary artery remodeling zone (CA rz). (F) Boxed region in C''' highlighting SM-MHC[low] cells (arrowheads) at the remodeling zone in comparison to the higher expression in cells surrounding a more mature coronary artery (arrow). (G) Histogram plotting SM-MHC expression shows that mural cells of the remodeling zone are SM-MHC[low] while those around mature arteries are SM-MHC[high] (n = 16 cells/region from 4 embryos). (H) Histogram plotting distance of SM-MHC[low] cells from the epicardium (n = 22 cells from 5 hearts). (I) Proposed model where an intermediate progenitor (IP) bridges epicardial cells (Epi) and coronary artery smooth muscle. Ao, aorta; pt, pulmonary trunk. Scale bars, 100 μm.

*Figure 1. continued on next page*

Figure 1. Continued

The following figure supplements are available for Figure 1:

**Figure supplement 1.** Morphology of SM-MHC[low] cells.

reporter system (*Zong et al., 2005*). Tbx18-Cre was selected because, in control experiments with other Cre reporters, recombination was induced in the large majority of epicardial cells (*Figure 2—figure supplement 1A*). The MADM system fluorescently labels Cre-expressing cells through a rare interchromosomal recombination event such that, when coupled with Tbx18-Cre, hearts contained isolated clusters of fluorescently labeled cells (*Figure 2—figure supplement 1B*). We analyzed tightly associated clusters (at least 100 µm from any other labeled cell) that contained epicardium, caSMCs, and other clonally related cells, the latter of which could be the progenitor population.

Initial experiments analyzing Tbx18-Cre, MADM clones based on cellular morphology and location revealed that clones containing epicardial cells and caSMCs always contained sister cells with long, thread-like processes that wrapped around coronary vessels (*Figure 2A*). This was in contrast to sister cells from clones that were adjacent to, but did not incorporate into, the caSMC layer, which were mostly located in the space between vessels instead of wrapping around them (*Figure 2B*). The morphology of the caSMC-associated sister cells was characteristic of vascular pericytes, which are mural cells that tightly associate with small blood vessels and regulate their development and function (*Armulik et al., 2011*). To investigate the presence of pericytes in caSMC clones, we validated the use of a set of markers that would allow us to identify pericytes and caSMCs in the developing heart. These were then used to analyze an additional set of clones. The following describes our marker selection criteria.

We found that whole mount immunostaining with PDGFRβ and SM-MHC appeared to distinguish between pericytes and caSMCs. CaSMCs were positive for SM-MHC and PDGFRβ and were around mature and developing arteries (*Figure 2C*). In contrast, PDGFRβ perivascular cells around microvessels were SM-MHC negative (*Figure 2C*). We also analyzed the expression of two additional smooth muscle markers, Smooth Muscle Alpha Actin (SMα and Calponin 1 (CNN1) (*Majesky, 2011*). Aside from a low level in some cardiomyocytes, SMα immunolabeling was only in larger, more proximal CAs, and not in PDGFRβ[+] cells around plexus capillaries (*Figure 2D*) or in SM-MHC[+] cells in smaller vessels and remodeling zones (*Figure 2E*). Thus, in the coronary system, SMα appeared to be induced later than SM-MHC since its expression domain was less extensive and only in more mature arteries. CNN1 was highly expressed in cardiomyocytes, but did not appear in intervening PDGFRβ[+]-cells (*Figure 2—figure supplement 2A*). Around large coronary arteries a subset of caSMCs expressed CNN1 (*Figure 2—figure supplement 2B*). Cells that surrounded plexus capillaries and were positive for PDGFRβ, but negative for SM-MHC, SMα, and CNN1 (*Figure 2C,D*, *Figure 2—figure supplement 2A*), displayed a morphology similar to vascular pericytes with long processes that wrap the endothelium (*Figure 2F*).

To investigate whether PDGFRβ[+] perivascular cells around microvessels could be cardiac pericytes (*Armulik et al., 2011*), we further characterized this population. Whole mount immunohistochemistry showed that PDGFRβ[+] cells surrounding plexus capillaries in the free walls of the developing heart ventricles were always closely apposed to endothelial cells in contrast to PDGFRα[+] fibroblasts, which were dispersed between the endothelium (*Figure 2G* and *Figure 2—figure supplement 3A,B*). PDGFRβ[+] cells were also embedded within a Collagen IV-containing basement membrane (*Figure 2H* and *Figure 2—figure supplement 3C,D*). These are critical attributes for pericyte identification (*Armulik et al., 2011*). They also all expressed the pericyte markers NG2 (*Figure 2I*) and Notch3 (*Figure 2J*) (*Liu et al., 2010*) (quantification shown in *Figure 2K*). PDGFRβ[+] cells were observed on the earliest coronary sprouts when the vessels first migrate directly beneath the epicardium and surrounded the entire plexus endothelium as it developed (*Figure 2—figure supplement 4*). Later, at the arterial remodeling zone where caSMCs first differentiate, 98 ± % of PDGFRβ[+] cells were Tbx18-Cre lineage labeled consistent with most arising from the epicardium (*Figure 2K*). In addition, PDGFRβ[+] cells were the most numerous Tbx18-Cre traced cell type in this location (*Figure 2L* and *Figure 2—figure supplement 5A*). The remaining 28% were predominately fibroblasts as defined by their expression of PDGFRα and variable proximity to the vessels (*Figure 2L* and *Figure 2—figure supplement 5B*). At time points after

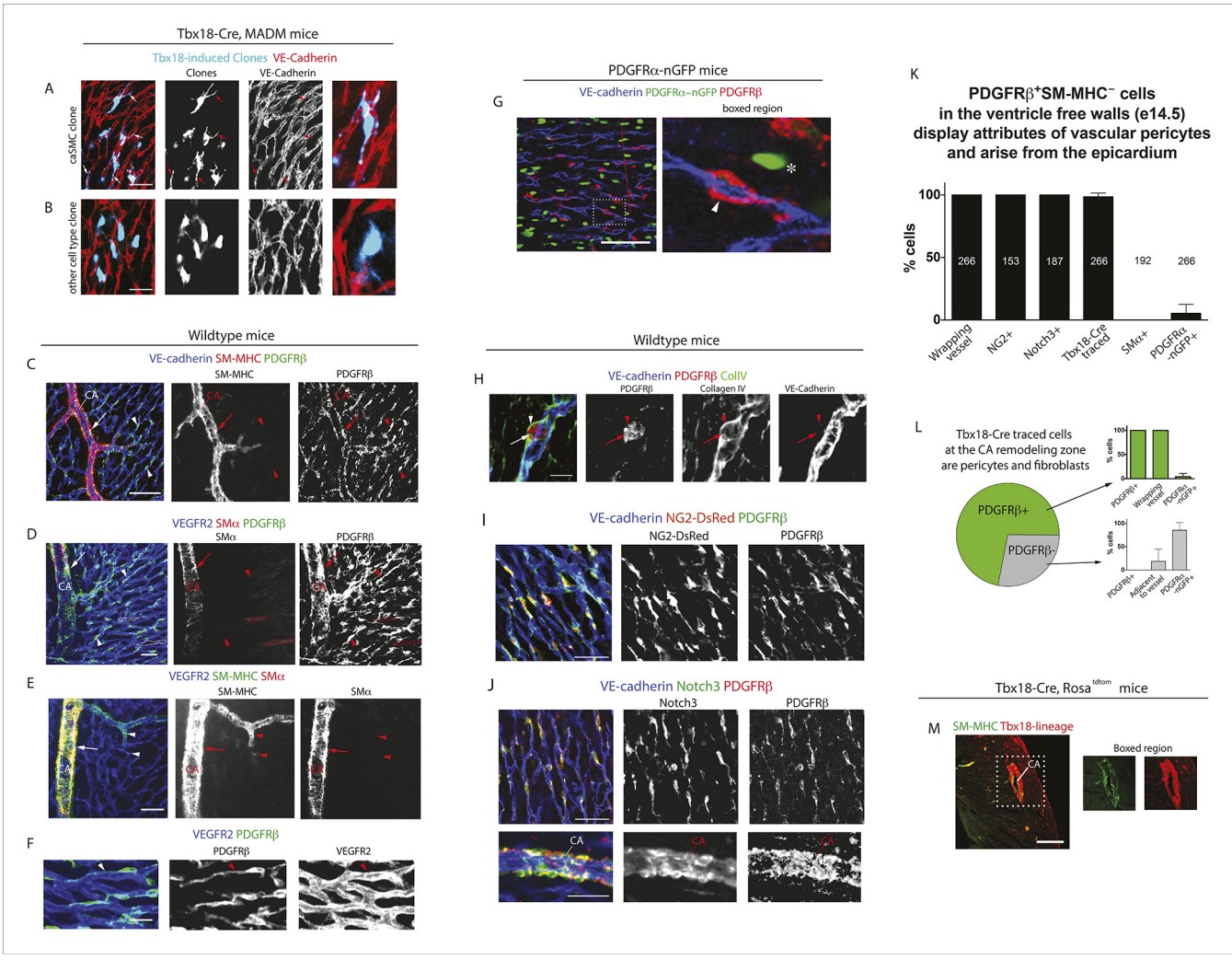

**Figure 2.** Characterization of epicardial-derived pericytes in the developing heart. (A–G) Whole mount confocal images of hearts immunostained with the indicated antibodies and/or fluorescent labels. (A and B) Images from Tbx18-Cre, MADM hearts show that coronary artery smooth muscle cell (caSMC) containing clones (blue) always include pericyte-like sister cells with long extended processes (arrows) that travel along VE-cadherin+ blood vessels (red) (A). In contrast, cells within clones not containing caSMCs (other cell type clone) are generally located in between vessels (B). High magnifications are shown on the right. (C–J) Mural cell characterization in hearts from mice of the indicated genotypes. (C and D) Smooth muscle and pericytes can be distinguished by immunolabeling for smooth muscle cell contractile proteins (SM-MHC and SMα) and PDGFRβ. (C) Smooth muscle surrounding coronary arteries (CA) is positive for SM-MHC and PDGFRβ (arrows) while pericytes only stain for PDGFRβ (arrowheads). (D) PDGFRβ+ pericytes are not labeled with SMα-specific antibodies (arrowheads). caSMCs around large arteries are positive for both markers (arrows). Some cardiomyocytes expression low levels of SMα (outlines). (E) SM-MHC is expressed in small and large arteries (arrowheads), while SMα only marks the caSMC coating around larger, more mature vessels (arrows). (F) PDGFRβ+ cells display a pericyte-like morphology with long processes that wrap around microvessels (arrowheads). (G) PDGFRβ immunostaining of PDGFRα-GFP hearts demonstrate that the two markers do not significantly overlap. PDGFRβ+ cells (red) wrap around the vessel (arrowhead), while PDGFRα+ cells usually exist in-between vessels (asterisk). (H) PDGFRβ+ cells (arrow) are embedded within a Collagen IV+ basement membrane (arrowhead). (I and J) PDGFRβ overlaps with NG2-DsRed labeling (I) and Notch3 immunostaining (J). (K) Quantification of marker expression and lineage labeling in PDGFRβ+ cells in the free walls of the developing heart ventricles. The number of cells analyzed are indicated. (L) PDGFRβ+ pericytes are the most numerous epicardial-derived cell type at the e14.5 coronary artery remodeling zone. 72% of Tbx18-Cre, Rosa^tdtomato lineage traced cells are pericytes (PDGFRβ+) (n = 14 hearts from 6 litters). The epicardial derived PDGFRβ- fraction contains mostly PDGFRα+ fibroblasts. (M) Tbx18-Cre, Rosa^tdtomato lineage tracing shows that the majority of caSMCs are epicardial derived. Scale bars, A and B, 20 μm; C–E, 50 μm; F and H, 10 μm; G, 50 μm; I and J, 100 μm; M, 200 μm.

The following figure supplements are available for Figure 2:

**Figure supplement 1.** Tbx18-Cre lineage tracing and clonal analysis.

**Figure supplement 2.** Calponin is not expressed in PDGFRβ+ perivascular cells that wrap microvessels.

*Figure 2. Continued*

**Figure supplement 3.** PDGFRβ⁺ perivascular cells are adjacent to vessels and within the basement membrane.

**Figure supplement 4.** Characterization of PDGFRβ perivascular cells in the developing heart.

**Figure supplement 5.** Epicardial-derived cells at the arterial remodeling zone are largely pericytes.

arteries are formed, much of the coronary smooth muscle was also lineage labeled with Tbx18-Cre (*Figure 2M*) in line with other studies showing an epicardial origin for these cells (*Cai et al., 2008b*; *Wilm et al., 2005*; *Zhou et al., 2008*). The epicardium is not thought to give rise to cells of the hematopoietic lineage, and, accordingly, PDGFRβ cells did not overlap with CD45 staining (data not shown). It must be noted that although eight criteria (PDGFRβ⁺Notch3⁺NG2⁺SM-MHC⁻SMα⁻PDGFRα⁻, wrapping microvessels, embedded within basement membrane) suggested that PDGFRβ⁺ perivascular cells are pericytes, we cannot exclude the possibility that the discovery of different markers will reveal this population to be a mix of pericytes and pericyte-like cells with different functions. However, based on current knowledge, the above described cell type will be referred to as pericytes below. In summary, given their marker expression, localization within the tissue, and cellular morphology, we defined PDGFRβ⁺ cells to be cardiac pericytes of the developing coronary vasculature. We also show that they are the most common epicardially-derived cell type at the arterial remodeling zone making them a prime candidate for caSMC progenitors.

To evaluate the likely clonality of the observed fluorescently labeled cell clusters, we mathematically modeled the process of clone formation using realistic parameters estimated from the imaging data of the e13.5 developing heart (see details in Materials and methods). In brief, we simulated the process of clone formation as a three dimensional Poisson process and estimated the underlying rate of clone generation that best fit the observed experimental data of ∼∼3.3 fluorescently labeled clusters per half heart (*Figure 3A*). For the estimated clone generation rate, we then calculated the fraction of simulated clusters that were truly clonal (not comprised of more than one overlapping clones) to evaluate the likelihood that the observed clusters are indeed clonal. This analysis revealed that ∼82% of the observed clusters were likely to derive from a single labeling event (*Figure 3B*). This same clonality rate was maintained when considered for hearts having a range of 1–6 clusters each (*Figure 3C*), which was the range observed in our e13.5 data set (mean value of 3.3 ± 1.6). We repeated the entire modeling process with different settings for the discrimination distance that defines clonal clusters (see details in Materials and methods) confirming the robustness of the estimated clonality rate (*Figure 3—figure supplement 1*). These observations provide evidence that greater than 80% of the observed clusters were truly clonal.

We next used PDGFRβ and SM-MHC to analyze Tbx18-Cre-derived clonal cell clusters isolated at e13.5 and 15.5, before and after caSMCs develop, respectively. This analysis revealed a clonal and spatial relationship between caSMCs and pericytes. At e13.5 before caSMCs are present, we obtained clones consisting of just epicardial cells and pericytes, the latter of which were always located directly beneath epicardial sister cells (*Figure 3D-e13.5, E*). At e15.5, clones with caSMCs always contained pericytes that filled the region of the myocardium between the epicardium and smooth muscle and were continuous with, and often touching, related caSMCs (*Figure 3D-e15.5, E*, *Figure 3—figure supplement 2A,B*, and *Videos 1,2*). As expected from the Tbx18-Cre lineage trace data (*Figure 2L*), pericytes were the most numerous cell type in these clones (*Figure 3—figure supplement 2C*). Tbx18-Cre labels cardiomycytes (*Cai et al., 2008b*), but these cells never appeared in clonal clusters with pericytes or smooth muscle. Together, the temporal sequence and spatial arrangement of Tbx18-Cre-derived clones suggested that pericytes are the intermediate differentiation step between epicardium and caSMCs.

We next tested whether the pericytes that were clonally related to caSMCs during embryogenesis persisted into adulthood or were depleted after development. Strikingly, adult Tbx18-Cre, MADM hearts frequently contained tightly packed clusters of lineage labeled cells consisting of pericytes and caSMCs (*Figure 3D*-adult) (n = 20). Pericytes at this stage exhibited even longer, thinner processes that tracked along the vessels for greater than 5–10 times the length of their cell bodies

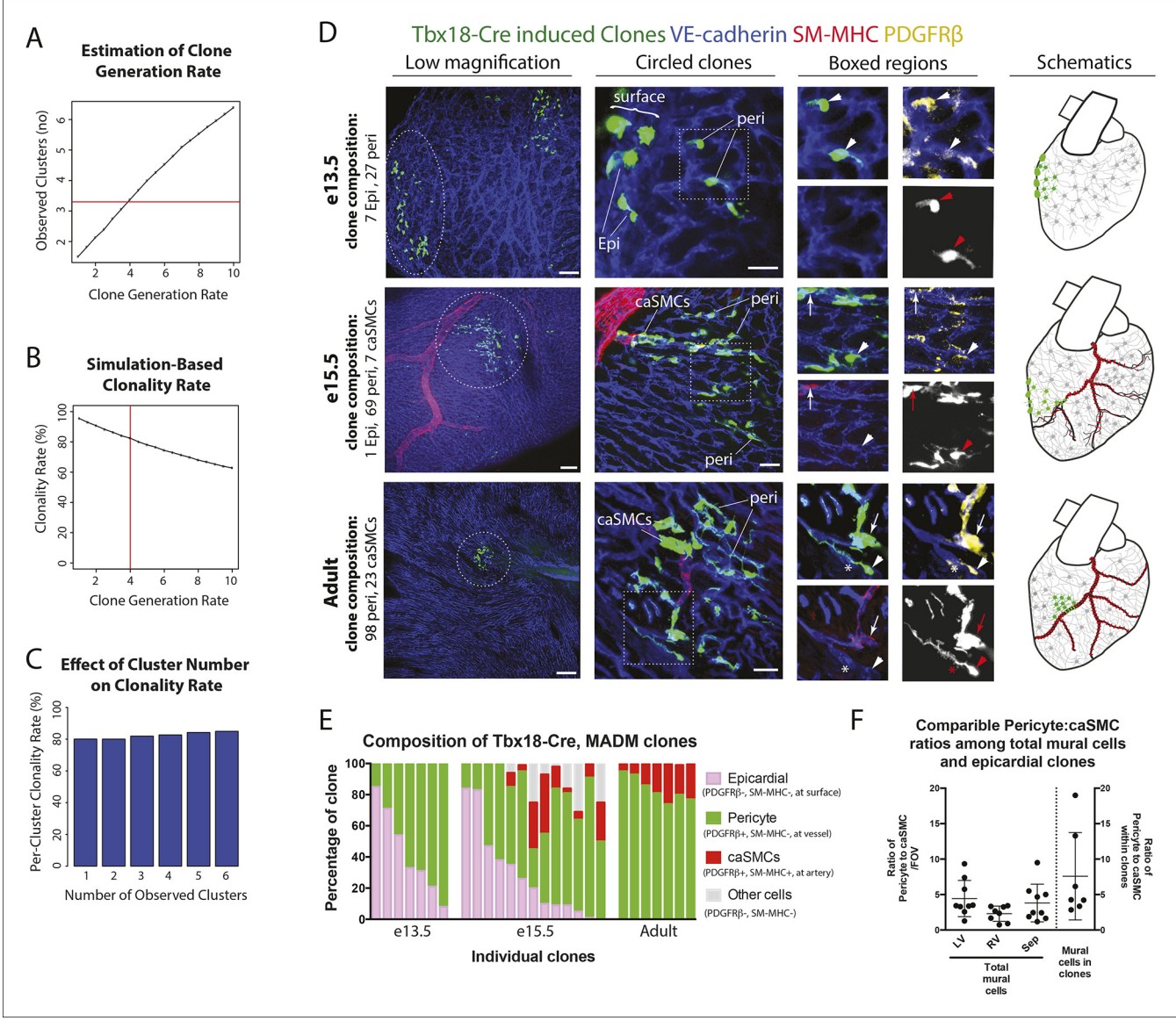

**Figure 3.** Coronary artery smooth muscle cells and pericytes are clonally related. (**A–C**) Simulation-Based Analysis of Clonality as outlined in Materials and methods. Mathematical modeling was performed to estimate the underlying clone generation rate that best fit the experimental data (denoted by the horizontal red line) (**A**), and to evaluate the corresponding overall average rate of clonality (denoted by the vertical red line) (**B**) as well as the clonality rates for simulated half heart regions with the designated numbers of observed clusters (**C**). (**D**) Confocal images of clones from indicated ages (e13.5, e15.5, and adult). Left panels are low magnification views of entire clones and middle panels are internal views of circled clones, which are near coronary arteries in e15.5 and adult. Boxed regions are separated channels as examples of marker expression with white showing clone label for morphology. Note that PDGFRβ staining is punctate while the clone label is uniform throughout the cell. Asterisks indicate long cellular processes in adult pericytes. Schematics of each are on the far right. (**E**) Graph showing cell types within individual clones. (**F**) Quantification of the percentage of pericytes and smooth muscle cells in clones and their ratios among total mural cells in adult hearts. caSMC, coronary artery smooth muscle cell; epi, epicardial cell; LV, left ventricle; peri, pericyte; RV, right ventricle; Sep, septum. Scale bars: left panels, 100 μm; middle panels, 20 μm.

The following figure supplements are available for Figure 3:

**Figure supplement 1.** Influence of discrimination distance parameter on estimated clonality rate.

**Figure supplement 2.** Additional examples of pericyte-coronary artery smooth muscle clones and quantification of cell location and number.

(*Figure 3D*-adult and *Figure 3—figure supplement 2D*). The fact that these cells were not dispersed, even five weeks after labeling, shows that pericytes and caSMCs do not migrate significantly

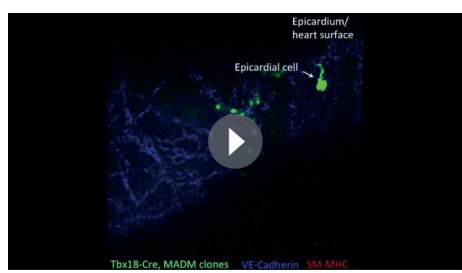

**Video 1.** Following an epicardial/pericyte/smooth muscle cell clone from the surface of the heart to a coronary artery deep in the myocardium. Movie showing single confocal Z-planes of the clone in *Figure 3A*-e15.5 going from the surface of the heart to deeper regions where a coronary artery is located (µms from the epicardium are indicated). A single epicardial cell is present at the surface while tightly clustered pericytes span the region between the surface and coronary artery where clonal cells are incorporated into the smooth muscle cell layer. VE-Cadherin (blue) labels the vasculature, SM-MHC (red) labels coronary artery smooth muscle, and the Tbx18-Cre, MADM lineage label is green.

once their positions are established during development. In addition, our analysis indicates that pericytes around capillaries that are clonally related to caSMCs remain and function as cardiac pericytes in the adult heart (*Figure 3E*). Within established adult clones, the ratio between the number of pericytes and caSMCs was comparable to ratios calculated for total mural cells from three different regions of the heart (*Figure 3F*). This is the expected result if the epicardial to pericyte pathway is a prominent source of caSMCs. In total, our clonal analysis suggests that epicardial-derived pericytes surround the entire coronary plexus during embryonic development, but that these cells differentiate into caSMCs if located around vessels that enlarge to become arteries during remodeling. These pericytes also surround capillaries in the adult heart and could potentially become new caSMC if arterial remodeling were induced at this stage.

## Lineage tracing pericytes in the developing heart

The above described composition and arrangement of epicardial-derived clones suggests that pericytes differentiate into caSMCs. However, direct lineage tracing of pericytes is required to confirm this sequence of events. NG2 and Notch3 are well-established pericyte markers in other tissues (*Armulik et al., 2011*; *Kofler et al., 2011*). In the myocardium of the developing ventricle free walls, NG2- and Notch3-positive cells on the microvasculature always expressed PDGFRβ and wrapped vessels (*Figure 2I–K*), which identified them as cardiac pericytes (*Figure 2*). NG2 and Notch3 also completely overlapped (n = 185 cells counted, data not shown). To test if pericytes differentiate into smooth muscle, we performed lineage tracing using two independent mouse lines, NG2-CreER (*Zhu et al., 2011*) and Notch3-CreER (*Fre et al., 2011*). In embryos containing either the NG2-CreER or Notch3-CreER allele coupled with the fluorescent Cre reporter gene $Rosa^{tdtomato}$, pericytes, but not caSMCs, were labeled by injecting tamoxifen at either e10.5 or e11.5. This restricts Cre activity to a time point after pericytes had formed but before caSMCs appear (*Figure 4A*). If caSMCs arise from pericytes, this strategy should result in lineage labeled caSMCs, i.e. tdtomato (*Figure 4B*).

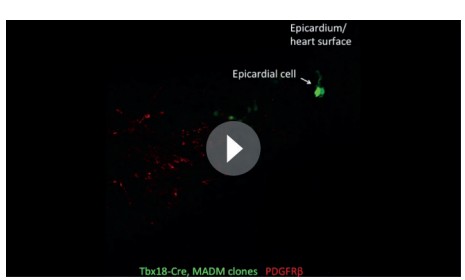

**Video 2.** Pericytes and smooth muscle clone cells express PDGFRβ. Clone in *Video 1* (*Figure 3A-e15.5*) showing the fluorescent channels for PDGFRβ in red and the Tbx18-Cre, MADM lineage label in green. Note that PDGFRβ is localized in a punctate pattern while the lineage label fills the entire cell so that only portions of the cell appear as double positive (yellow).

These lineage tracing experiments resulted in labeled caSMCs with both Cre drivers. Importantly, no labeling was detected in the absence of tamoxifen (*Figure 4C* and data not shown). Quantifying the proportion of lineage labeled pericytes in the compact myocardium indicated that Cre recombination efficiency in $PDGFR\beta^{+}$-$SM\text{-}MHC^{-}$ perivascular cells (pericytes) was 21% and 38% with NG2-CreER and Notch3-CreER, respectively (*Figure 4D*). $SM\text{-}MHC^{+}$ caSMCs also expressed the tdtomato lineage label in both NG2-CreER (*Figure 4E,F*) and Notch3-CreER (*Figure 4G,H*) strains. Similar percentages of labeling were seen in pericytes and caSMC (quantified in the compact myocardium) consistent with low recombination efficiencies in pericytes accounting for the incomplete caSMC tracing (*Figure 4D*). However, our analysis does

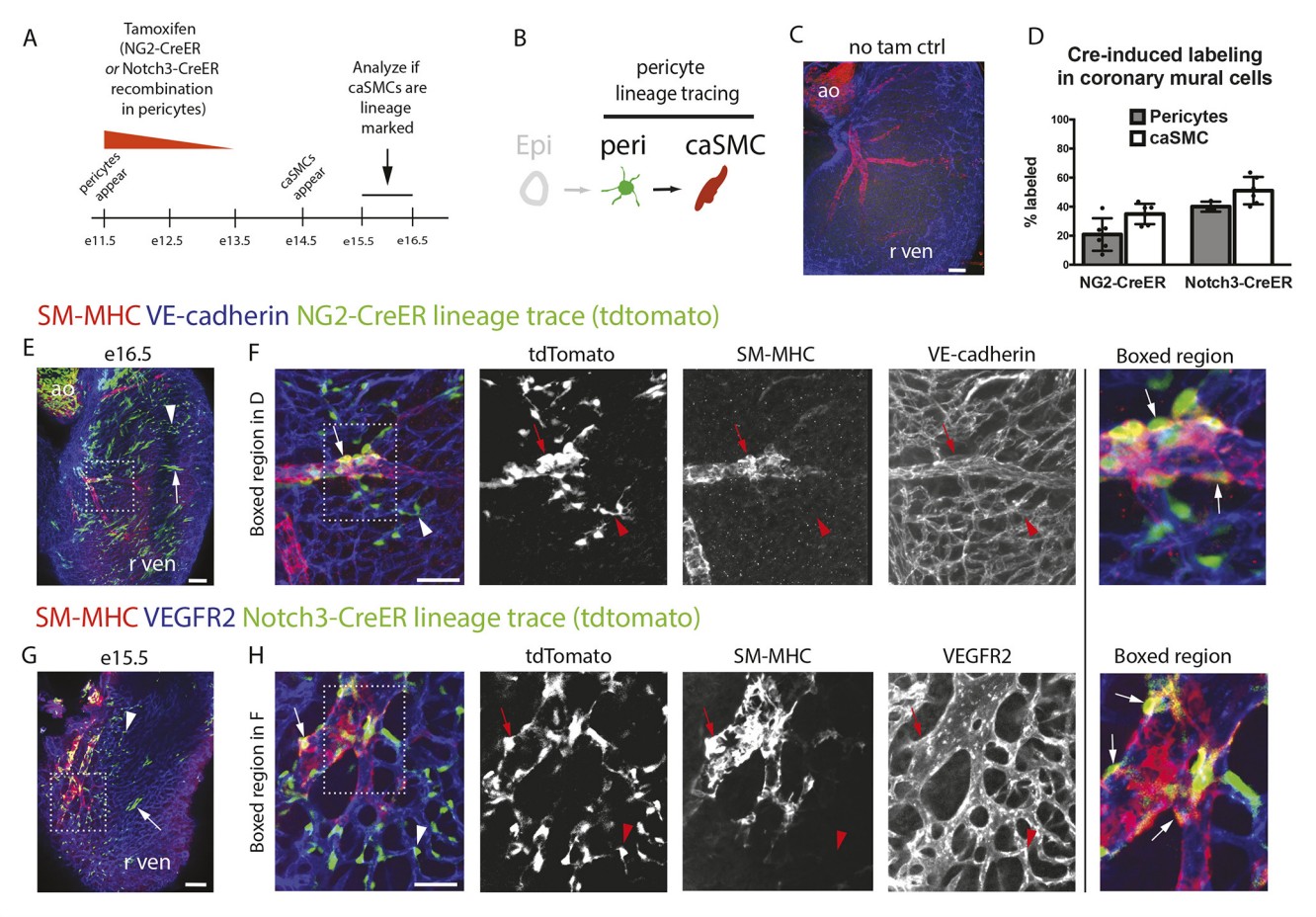

**Figure 4.** Pericytes differentiate into coronary artery smooth muscle. (**A** and **B**) Schematics describing the experimental design for cardiac pericyte lineage tracing (**A**) and part of the differentiation pathway being interrogated (**B**). (**C**) No recombination occurs in NG2-CreER, Rosa$^{tdtomato}$ animals in the absence of tamoxifen (tam). (**D**) Quantification of Cre labeling (i.e. recombination efficiency) in NG2$^+$Notch3$^+$ pericytes alongside levels of smooth muscle lineage labeling. (**E**) E11.5 dosing of NG2-CreER induces lineage labeling (green) in pericytes (arrowhead), smooth muscle, and some cardiomyocytes (arrow). (**F**) Boxed region in **E** showing lineage labeled pericytes (green, arrowheads) and coronary artery smooth muscle (yellow, arrows) (n = 10 hearts from 3 litters). Endothelial cells are in blue (VE-cadherin$^+$). Right panel is boxed region in far left panel. (**G**) Labeled pericytes (arrowhead), smooth muscle, and rare cardiomyocytes (arrow) in Notch3-CreER lineage trace. (**H**) Boxed region in **F** showing lineage labeled pericytes (green, arrowheads) and coronary artery smooth muscle (yellow, arrows) (n = 11 hearts from 2 litters). Endothelial cells are in blue (VEGFR2$^+$). Right panel is boxed region in far left panel. Ao, aorta; caSMC, coronary artery smooth muscle cell; epi, epicardium; r ven, right ventricle, Scale bars: C, E and G,100 μm; F and H 50 μm.

not exclude the possibility of an additional source, particularly in regions not analyzed, i.e. the ventricular septum. One potential confounding factor is that both transgenes exhibited sporadic labeling in cardiomyocytes (*Ozerdem et al., 2001*), which was more rare in Notch3-CreER (*Figure 4E,G*). However, control experiments using Myh6-CreER showed that cardiomyocyte lineage tracing never labeled caSMCs (data not shown). Finally, NG2 and Notch3 were expressed in PDGFRβ perivascular cells that were characterized as cardiac pericytes (*Figure 2*), but, as stated above, we cannot exclude the possibility that these experiments traced pericyte-like SMC progenitors sharing attributes with traditional pericytes, specifically PDGFRβ$^+$Notch3$^+$NG2$^+$SM-MHC$^-$SMα$^-$PDGFRα$^-$, wrapping microvessels, and embedded within the basement membrane. In summary, lineage tracing data from two independent Cre lines support a model where pericytes differentiate into caSMCs.

## Pericytes in PDGFRβ-deficient hearts

To gain further evidence that pericytes were caSMC progenitors, we analyzed mutants with deficient caSMC layers and asked if this phenotype was correlated with defects in pericytes at the arterial

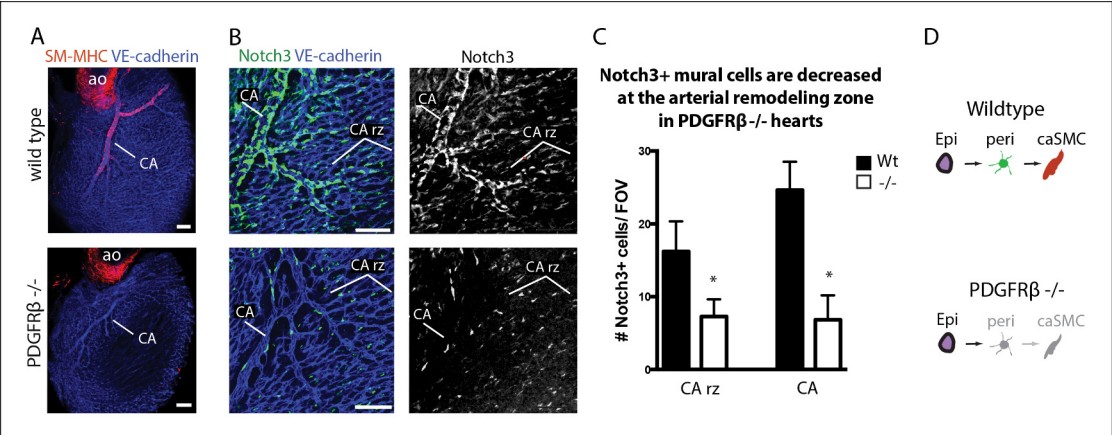

**Figure 5.** Coronary artery smooth muscle and pericytes are decreased in PDGFRβ-null mice. (**A**) Absence of SM-MHC[+] smooth muscle around coronary arteries (CA) in PDGFRβ knockout hearts. (**B**) Notch3[+] mural cells (green) are decreased at the coronary artery remodeling zone (CA rz) in PDGFRβ-deficient hearts (n = 8 from 4 litters). (**C**) Quantification of pericyte numbers per field of view (FOV) (wild type, n = 7 hearts; mutant, n = 5). Error bars are s. d.; *p≤0.05. (**D**) Schematic demonstrating the hypothesized epicardial to smooth muscle differentiation pathway and how it is affected in PDGFRβ-null mice. Greyed cells are reduced or absent. Ao, aorta; CA rz, coronary artery remodeling zone; caSMC, coronary artery smooth muscle cell; epi, epicardium; r ven, right ventricle. Scale bars: A, 100 μm; B, 50 μm.

maturation zone. PDGFRβ-null mice have reduced caSMCs (*Figure 5A*) (*Hellström et al., 1999*; *Smith et al., 2011*); however, they undergo vascular remodeling exhibiting a recognizable arterial remodeling zone and CA (*Figure 5B*). Analyzing the total number of Notch3[+] pericytes at the remodeling zone revealed that they were significantly decreased (*Figure 5B,C*). Pericytes and caSMCs were not completely gone (*Figure 5B,C*) suggesting the presence of a very inefficient compensatory mechanism. The coincident decrease in pericytes and caSMC support a model where pericytes are progenitors that are required for smooth muscle formation, and that PDGFRβ functions at the surface to induce the first transition of the pathway from epicardial cell to pericyte (*Figure 5D*).

## Notch3 functions in the pericyte to caSMC transition

To identify molecular regulators of the pericyte to caSMC differentiation step, we characterized the expression of mural cell markers at the arterial remodeling zone. Among those investigated, Notch3 was upregulated in pericytes at the region where SM-MHC protein expression was initiated (*Figure 6A,B*). In fact, the arterial remodeling zone can be identified based on Notch3 staining intensity without visualizing vessel structure (*Figure 6A*). This pattern was in contrast to PDGFRβ, which was uniformly expressed in mural cells throughout the vasculature (*Figure 6A,B*). Jagged-1 has been shown to be a major ligand for Notch3 in other organs, where it is expressed in arterial endothelium and stimulates smooth muscle differentiation including inducing the expression of SM-MHC (*Briot et al., 2014*; *Doi et al., 2006*; *High et al., 2008*; *Hofmann et al., 2012*; *Jin et al., 2008*; *Kofler et al., 2011*; *Manderfield et al., 2012*; *Yang and Proweller, 2011*). We found high levels of Jagged-1 to be specifically expressed at arterial remodeling zones where SM-MHC[+] cells were developing and in CAs (*Figure 6C* and *Figure 6—figure supplement 1*). Given that Jagged-1 has previously been identified as a shear stress-induced molecule in vitro (*McCormick et al., 2001*), we investigated whether it could be regulated by blood flow in the heart. Coronary vessels are initially unperfused, but begin to receive blood flow after they attach to the aorta at e13.5 (*Chen et al., 2014a*). This time point correlated with the onset of robust Jagged-1 expression in endothelial cells of vessels directly downstream of the attachment site (*Figure 6D–WT* and *Figure 6—figure supplement 1*). We analyzed a model of delayed CA stem attachment to the aorta, Isl1 heterozygosity (*Cai et al., 2008a*), which postpones the initiation of blood flow (*Chen et al., 2014a*). Immunostaining Isl1 mutant hearts revealed that Jagged-1 was only upregulated in arterial remodeling zones of e13.5 hearts that had formed CA stems on the aorta to receive blood flow (*Figure 6D,E*). Isl1 mutant

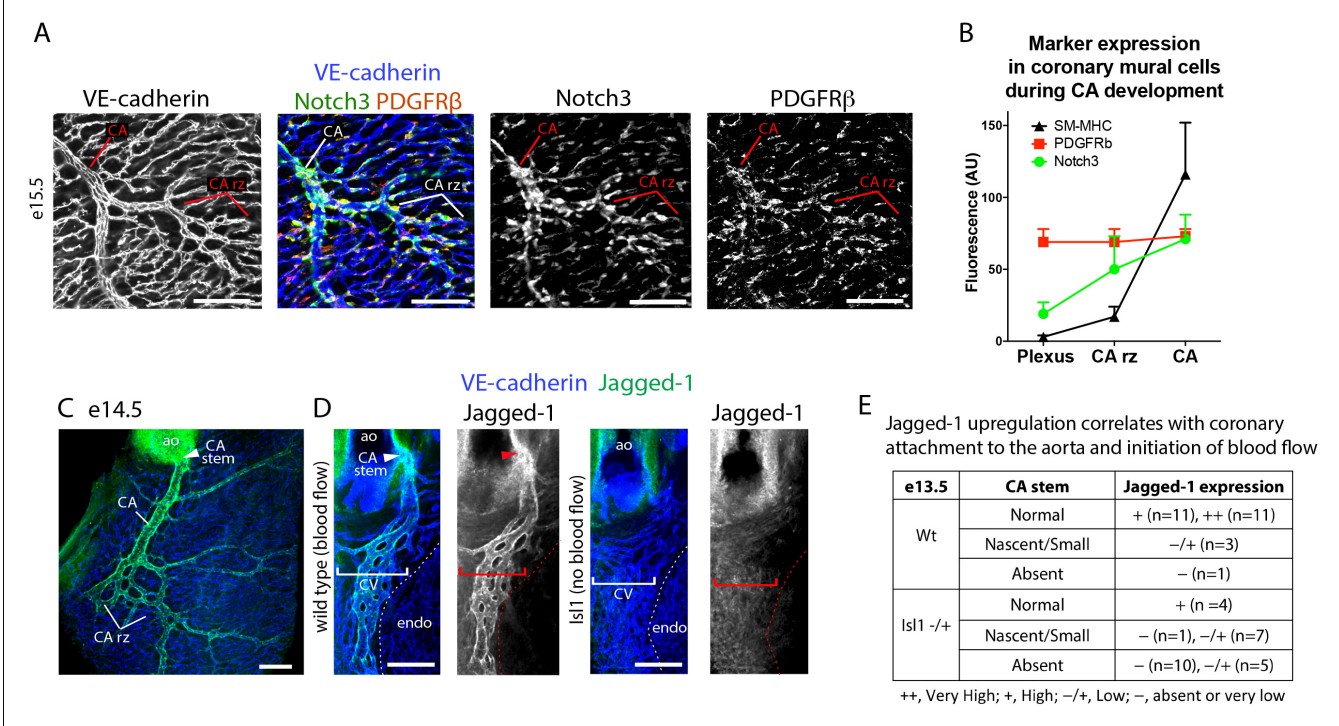

**Figure 6.** Notch3 and Jagged-1 expression at the arterial remodeling zone. (**A** and **B**) Mural cells around coronary vessels increase Notch3 protein expression at the coronary artery remodeling zone (CA rz) while PDGFRβ levels remain the same. (**A**) Confocal image of a representative remodeling zone. (**B**) Quantification of marker expression. Error bars are s.d. (**C** and **D**) Confocal images immunostained for VE-cadherin (blue) and Jagged-1 (green). (**C**) Jagged-1 is specifically expressed in coronary arteries (CA) and the CA rz after attachment to the aorta (ao) and induction of blood flow. (**D**) Jagged-1 is expressed in coronary vessels at e13.5 soon after aortic attachment and CA stem formation, but not in Isl1 mutant littermates with delays in attachment and arterial blood flow. (**E**) Table of Jagged-1 protein expression in wild type (Wt) and Isl1 mutants. Scale bars: 100 μm

The following figure supplements are available for Figure 6:

**Figure supplement 1.** Characterization of Jagged-1 expression during coronary artery development.

hearts also failed to upregulate Notch3 in pericytes suggesting that this change also requires signals downstream of blood flow (n = 9, data not shown). In summary, Notch3 is upregulated in pericytes at the arterial remodeling zone while its ligand Jagged-1 is induced in endothelial cells in the same region following the initiation of blood flow suggesting that this receptor-ligand pair could trigger caSMC differentiation in coronary pericytes.

We next analyzed Notch3-deficient hearts to investigate whether this signaling pathway could be involved in the pericyte to caSMC transition. CAs in Notch3-null mice displayed significantly reduced levels of SM-MHC when compared to controls at two time points examined, e15.5 and e17.5 (*Figure 7A,B* and data not shown). SMα was also reduced (*Figure 7C*). In contrast, CA coverage by PDGFRβ⁻ cells was similar to wild type counterparts (*Figure 7D*), and PDGFRβ staining revealed that pericytes were still present in normal numbers in the absence of Notch3 (*Figure 7D,E*). We next used EdU labeling to mark cycling cells in wildtype and knockout hearts. Quantification of EdU⁺ mural cells (PDGFRβ⁺) showed no difference between the two genotypes (*Figure 7F*). Thus, epicardial to pericyte differentiation and arterial coverage is not severely affected, but caSMC maturation, specifically the induction of contractile proteins (SM-MHC and SMα), is disrupted (*Figure 7G*). These data show that coronary vessels upregulate Jagged-1 after attaching to the aorta to receive arterial blood flow, and that Notch3 is required for pericyte to caSMCs differentiation, possibly in response to Jagged-1 expression at the arterial remodeling zone.

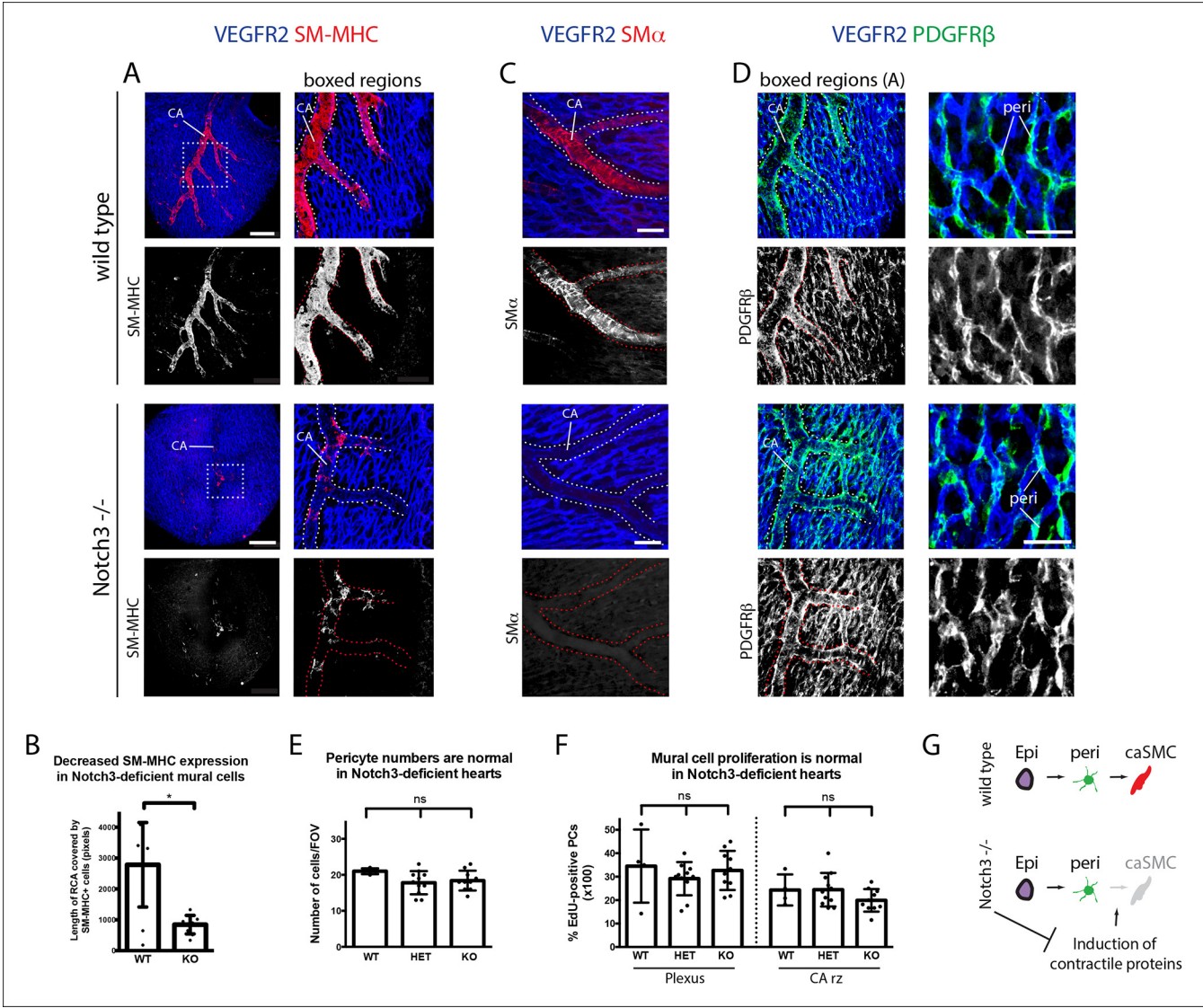

**Figure 7.** Notch3 is required for coronary artery smooth muscle development. (**A**) SM-MHC⁺ coronary artery smooth muscle cell (caSMCs) are significantly reduced in Notch3-null hearts although coronary artery (CA) caliber (dotted lines) is comparable. (**B**) Quantification of caSMC coverage in Notch3-deficient hearts where dots are individual samples and error bars are s.d. *p≤0.05. (**C**) SMα protein expression is reduced on arteries from Notch3-deficient hearts. (**D**) PDGFRβ⁺ cells cover CAs in both wild type and knockout, and pericytes (peri) are not significantly reduced. (**E**) Quantification of pericyte numbers in Notch3-deficient hearts. (**F**) Quantification of mural cell proliferation in Notch3-deficient hearts at the capillary plexus and CA remodeling zone (CA rz). (**G**) Schematic demonstrating the hypothesized epicardial to smooth muscle differentiation pathway and how it is affected in the absence of Notch3. Greyed cells are reduced. Scale bars: A, 100 μm; C, 50 μm; D, 25 μm.

## NG2⁺ and Notch3⁺ cells are smooth muscle progenitors in the kidney

We next investigated whether the pericyte to smooth muscle transition occurs in the kidney, another organ who's internal arteries receive smooth muscle from the surface mesothelium (*Rinkevich et al., 2012*). Characterization of smooth muscle development in the kidney using whole mount confocal microscopy showed that SM-MHC⁺ cells first appear at e14 on developing intralobular arteries (*Figure 8A*). As in the heart, NG2 also labels pericytes in the developing kidney (*Lin et al., 2008*). We therefore lineage traced either NG2- or Notch3-positive cells by inducing labeling of the Rosa^tdtomato Cre reporter before smooth muscle appears (*Figure 8B*). Both approaches resulted in robust lineage labeling of arterial smooth muscle cells in the kidney at e15.5 as well as other perivascular cells including those within the glomerulus (*Figure 8C* and data not shown). Although understanding the precise cellular pathway from mesothelium to smooth muscle

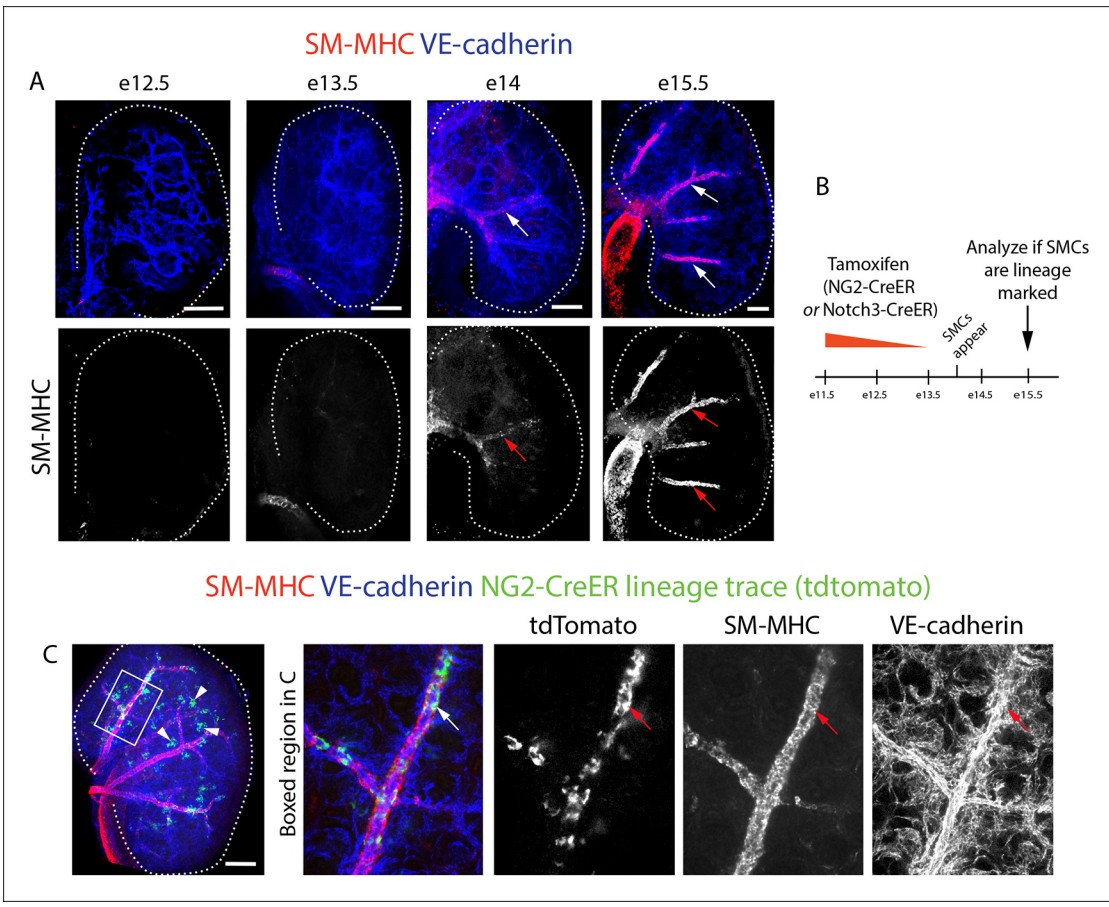

**Figure 8.** NG2[+] and Notch3[+] cells differentiate into smooth muscle cells in the kidney. (**A**) Whole mount confocal imaging of embryonic kidneys (outlined with dotted lines) from the indicated ages immunostained for SM-MHC and VE-cadherin. Mature smooth muscle differentiation is detected at e14. (**B**) Schematic describing lineage tracing experimental design. (**C**) e11.5 dosing of NG2-CreER, Rosa^tdtomato animals induces labeling (green) in smooth muscle (red, arrows)  (n = 11 kidneys from 2 litters). Cells within the glomerulus are also labeled (arrowheads). Scale bars: 100 μm.

requires further study, these data suggest that, similar to the heart, pericytes form an intermediate stage during this differentiation process in the kidney.

## Discussion

Due to the cells' proximity and similar marker expression, vascular biologists have long wondered if pericytes and smooth muscle interconvert (*Armulik et al., 2011*; *Cappellari et al., 2013*; *Majesky, 2011*). Here, we show evidence, for the first time, that pericytes differentiate into smooth muscle cells during coronary artery development in the mouse heart and developing kidney. Previous identification of the epicardial-derived caSMC progenitor had been hampered by the fact that multiple developmental pathways occur downstream of epicardial differentiation. We have overcome this hurdle by using clonal analysis to label differentiation steps downstream of a single epicardial cell. This single cell tracing analyzes whether cells are clonally related eliminating one caveat of population level Cre labeling experiments where a mistake in the cell types expressing Cre can produce misleading results. Our clonal analysis of epicardial-derived cells identified cells with the molecular marker and morphological profile of vascular pericytes as being lineage related to caSMCs. The temporal presence and position of pericytes in epicardial-derived clones suggested that pericytes were the intermediate differentiation step between the epicardium and smooth muscle. This sequence of events was confirmed by direct pericyte lineage tracing using two independent pericyte

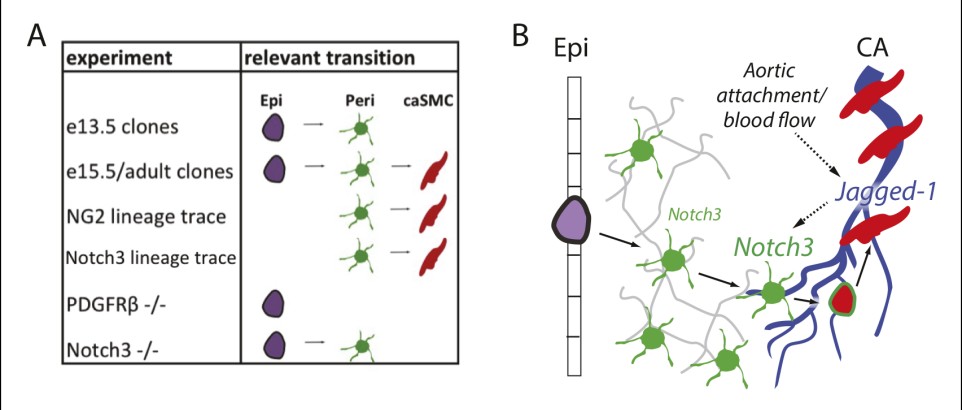

**Figure 9.** Model and summary. (**A**) Different parts of the hypothesized epicardial to caSMC pathway were dissected using the indicated experiments. (**B**) Working model for caSMC differentiation. CA, coronary artery; caSMC, coronary artery smooth muscle cell; Epi, epicardium; Peri, pericytes.

Cre lines. In addition, pericytes were the most numerous epicardial-derived cell type at the CA remodeling zone where they wrapped developing arteries, a location that would allow them to respond to arterial maturation signals. Thus, we used complimentary lineage analysis experiments consisting of clonal and population-based tracing to show that pericytes (or pericyte-like cells) travel along blood vessels as they enter the myocardium and function as progenitors for caSMCs.

Since pericytes wrap all small coronary vessels, we explored the signals that trigger their differentiation into caSMCs specifically at CA remodeling zones. We found that upon attachment to the aorta and establishment of blood flow, coronary endothelial cells downstream of the attachment site, which presumably receive the strongest blood flow, induce Jagged-1. Jagged-1 was not robustly expressed in a model of delayed aortic attachment and delayed initiation of blood flow. The Jagged-1 ligand Notch3 was upregulated in pericytes at the arterial remodeling zone, and caSMCs failed to differentiate in Notch3-null hearts. A delay in smooth muscle differentiation in the absence of Notch3 and Jagged-1 has been observed in other organ systems (*Briot et al., 2014*; *Doi et al., 2006*; *High et al., 2008*; *Hofmann et al., 2012*; *Jin et al., 2008*; *Manderfield et al., 2012*; *Yang and Proweller, 2011*), however nothing is known about their roles in caSMC differentiation. Furthermore, due to the previous lack of knowledge of the lineage relationship between pericytes and smooth muscle cells, we are the first to show the importance of Notch3 signaling in stimulating the pericyte to smooth muscle transition. Jagged-1 and Notch signaling has been reported to be upregulated by sheer stress in vitro(*McCormick et al., 2001*; *Theodoris et al., 2015*), and we provide evidence that this occurs in vivo following coronary plexus attachment to the aorta. Together, our data support a working model (*Figure 9*) where epicardial cells differentiate into Notch3$^{low}$ pericytes that cover the entire coronary plexus as it populates the myocardium. Then, the pericytes located on plexus vessels that receive blood flow-induced Jagged-1 upregulate Notch3 and differentiate into caSMCs. Thus, blood flow initiation is correlated with molecular changes that mark the site of arterial development in the coronary vascular plexus and may stimulate vessel remodeling and pericyte differentiation into smooth muscle.

Our observations on the pericyte to smooth muscle transition are consistent with studies analyzing the timing of epicardial differentiation and its emergence into the myocardium. Studies dissecting the role of PDGFRs, TCF21, and Myocardin-Related Transcription Factors have provided evidence that the decision to differentiate into either the cardiac fibroblast or caSMC lineage occurs in epicardial cells at the heart's surface (*Acharya et al., 2012*; *Braitsch et al., 2012*; *Mellgren, et al., 2008*; *Smith et al., 2011*; *Trembley et al., 2015*). In addition, progression down the caSMC lineage occurs early in development, mostly between e10.5 and 12.5 (*Wei et al., 2015*). We show that endothelial cells migrating directly beneath the epicardium at this time point acquire pericytes and pericytes within caSMC containing clones are usually located near the heart's surface as well as deeper layers. Thus, migrating coronary vessels could trigger adjacent epicardial cells to differentiate into pericytes at the surface and support their movement into the myocardium

along with invading endothelial cells. Then, the subset of these pericytes that traveled to arterializing vessels receives signals to become caSMCs. Clones also generally segregated between those containing pericytes/caSMCs and those with other cell types not positive for our cell type specific markers, likely fibroblasts, the latter of which did not integrate into the smooth muscle layer even when directly adjacent to the artery. Together, these findings provide strong evidence that two different epicardial derived pathways diverge at the surface of the heart where the pericyte/caSMC lineage travels along vessels as they invade the myocardium to provide the different mural cells of the coronary vasculature.

Analyzing pericyte-caSMC clones in adults showed that caSMC related pericytes persist after development is complete. Labeled cells in the adult heart were found in surprisingly tight clusters. This suggests that embryonic pericytes within caSMC clones were not merely an intermediate cell type that is depleted during development, but that cells not differentiating remain and function as cardiac pericytes throughout life. The tight clustering in adult hearts also indicate that pericytes travel very little along a vessel once they establish their location during development, at least in this organ. Thus, the embryonic heart appears to have developed an efficient method of distributing smooth muscle progenitors, cells that our data suggests maintain their cell-type specific function as pericytes at vessels that do not receive arterial blood flow and differentiation signals.

Studies in the adult heart and other adult tissues report that perivascular cells are a heterogeneous population with mesenchymal stem cell properties that can contribute to tissue fibrosis (*Chen et al., 2015*; *Ding et al., 2012*; *Kramann et al., 2014*; *Dulauroy et al., 2012*; *Stallcup, 2013*). Because a large number of caSMCs arise from the epicardium, we characterized epicardial-derived cells at the coronary artery remodeling zone. These were found to be almost exclusively either PDGFRβ$^+$Notch3$^+$NG2$^+$PDGFRα$^-$ (pericytes) or PDGFRβ$^-$Notch3$^-$PDGFRα$^+$ (fibroblasts) suggesting less heterogeneity in the epicardial-derived cellular compartment at this age. This could be due to development being an early and protected stage in the animal's life or that the cellular milieu is more complex in the adult when immunity is active. Additional complexity is likely induced in tissue injury models. Although lineage tracing shows that pericytes are a large source, our data does not exclude other pathways to caSMC development. In fact, developmental studies have show that an unidentified compensatory progenitor can provide caSMC if the epicardium is inhibited, although this source predisposes the arteries to disease (*Smith et al., 2011*; *Wei et al., 2015*). Regardless, the description of pericytes as progenitors and their presence in the adult identifies the possibility that they could be utilized to produce new caSMCs during CA regeneration. Identifying the cellular pathway between the epicardium and smooth muscle is therefore important, particularly since it is known that epicardial cells do not enter the adult heart, even after myocardial infarction (*Zhou et al., 2011*). Our data showing that pericytes are coronary artery smooth muscle progenitors and that they persist in the adult present the possibility that pericytes could be targeted without relying on the original progenitors (epicardial cells) being transported to form new collateral arteries.

Future experiments will investigate whether diverted arterial blood flow following coronary artery stenosis and/or blockage induces Notch signaling and differentiation of pericytes into caSMCs as part of the process that forms collateral vessels. If so, it will be important to ascertain whether collateral formation can be made more efficient by stimulating Notch activity. Our data suggest a similar pericyte to smooth muscle pathway in the kidney. However, it will be important to investigate whether smooth muscle in all mesothelial covered abdominal and thoracic organs and other tissues pass through a similar pericyte intermediate as it differentiates into organ-specific smooth muscle (*Rinkevich et al., 2012*).

## Materials and methods

### Animals

All animal experiments were performed according to protocols approved by the Stanford University Institutional Animal Care and Use Committee (IACUC). Mouse strains used: CD1 mice were used for wild type analysis and obtained from Charles River (San Diego, California). NG2-CreER (Jax strain: B6.Cg-Tg (Cspg4-cre/Esr1*)BAkik/J). (*Zhu et al., 2011*), NG2-DsRed (Jax strain: (cspg4-DsRed.T1) 1Akik/J), PDGFRα (Jax strain: B6.129S4-Pdgfratm11(EGFP)Sor/J), α-MHC-Cre (Jax strain: Tg(Myh6-

cre)1Jmk/J), MADM TG/TG (Jax strain: Iis2tm2(ACTB-tdTomato,-EGFP)Luo/J), MADM GT/GT (Jax strain: Iis2tm1(ACTB-EGFP,-tdTomato)Luo/J), Rosa-tdTomato (Jax strain: B6.Cg-Gt(ROSA)26Sortm9 (CAG-tdTomato)Hze/J), PDGFRβ flox (Jax strain: 129S4/SvJae-Pdgfrbtm11Sor/J), and Rosa-mTmG (Jax strain: Gt (ROSA)26Sortm4(ACTB-tdTomato,-EGFP)Luo/J) were all obtained from Jackson Laboratories. Conditional PDGFRβ flox animals were converted to full knockouts by crossing with HPRT-Cre females. *Isl1* heterozygous (*Cai et al., 2008a*), Tbx18-Cre (*Cai et al., 2008b*), Notch3-CreER (*Fre et al., 2011*) and Notch3 (*Krebs et al., 2003*) have been previously described.

## Immunohistochemistry and imaging

Staged embryonic hearts were obtained by timed pregnancies (morning plug designated e 0.5) and were dissected and fixed in 4% paraformaldehyde, washed and stored at 4°C in phosphate buffered saline (PBS). Whole mount fluorescence microscopy was performed on intact hearts. Staining was performed in 1.5 ml tubes subjected to constant rotation. Primary antibodies were diluted in blocking solution (5% goat serum, 0.5% TritonX-100 in PBS) and incubated with tissues overnight at 4°C. Tissues were then washed with PBT (PBS with 0.5% TritonX-100) four times for one hour before another overnight incubation with secondary antibodies diluted in blocking solution. Specimens were then washed again, placed in Vectashield (Vector Labs #H-1000), and imaged using an inverted Zeiss LSM-700 confocal microscope. Images were digitally captured and processed using Zeiss Zen software (2011).

The following primary antibodies were used: VE-cadherin (BD Biosciences, 550548, 1:100), PDGFRβ (R&D Systems, #BAF1042 1:50; eBioscience 14-1402-81, 1:100), Notch3 (Santa Cruz Biotechnology #M-134 1:100), SM-MHC (Biomedical Technologies, BT-562, 1:300), VEGFR2 (R&D Systems #AF644, 1:125); Jagged-1 (R&D Systems AF599, 1:125); SMα Quartzy #D00019 1:300); Calponin (Sigma-Aldrich #C6047, 1:250) Secondary reagents were Alexa Fluor conjugated antibodies (405, 488, 555, 637) from Life technologies used at 1:250or streptavidin conjugates (Life technologies #S21374) used at 1:500.

## Quantification (distance, fluorescence, and cell numbers) in whole mount preparations

In *Figure 1H*, distance of SM-MHC[low] cells from the epicardium was measured from Z-plane views of whole mount confocal images using Zeiss Zen software. Fluorescence intensity values were calculated from single Z-planes using the same software package. For *Figure 1G*, individual cells in single Z-planes were encircled and intensity values in the SM-MHC channel were recorded for mural cells surrounding the capillary plexus (n = 16), remodeling zone (n = 16), and mature arteries (n = 16) from 4 different hearts each. For *Figure 6B*, values in the PDGFRβ n = 9 cells/region from 3 hearts), Notch3n = 21 cells/region from 7 hearts), and SM-MHC (n = 15 cells/region from 3 hearts) channels were recorded. For *Figure 2—figure supplement 2B*, localization of PDGFRα+ and PDGFRβ+ cells was measured using the profile option where fluorescent intensities are graphed along a line drawn across the XY-plane of a confocal image. Quantification of PDGFRβ, NG2, and Notch3 overlap (mentioned in the text and shown in *Figure 2F,G*) was performed by randomly designating a field of view, encircling all the cells positive for either PDGFRβ or NG2, and counting the number of those circled also positive for the additional markers. Number of cells analyzed is stated in *Figure 2H* counted from 3–4 hearts per marker, each from multiple litters.

## Clonal analysis

Mice were bred so that embryos receive each of three alleles: Tbx18-Cre, a MADM GT cassette, and a TG cassette (*Zong et al., 2005*). Embryonic hearts were isolated at e13.5, e15.5 and 3–5 weeks of age and immunostained with antibodies for VE-cadherin, SM-MHC and PDGFRβ, and imaged as described above. For adults, 50 μm cryosectioning was performed and the sections were stained with the staining protocol as described above. Clusters of labeled cells were considered clonal if they were clearly distinct and at least 100 μm away from other labeled cells. For e13.5, e15.5 and adult clones a total of 12, 13 and 7 caSMC clones were quantified, respectively. Cell identities were assigned by the immunostaining criteria described in the main text and assessed by two individual researchers in the laboratory. Depth below the epicardium was analyzed using measurement tools included in the Zeiss Zen software (*Figure 3—figure supplement 1A*). Quantification of pericytes,

caSMCs and other cell types included all e15.5 clonal cells from 14 clones where no cells were excluded. In total, n values were 91 for epicardial cells, 249 for pericytes, 37 for caSMCs, 150 for other cell types.

## Detailed description of simulation-based analysis to evaluate clonality

To assess the expected clonality rate for the observed fluorescently labeled cell clusters, we mathematically modeled the process of clone formation in three dimensions based on parameters estimated from the imaging data of the e13.5 developing heart. The surface area of the heart was approximated as that of a sphere with a radius of 315 µm, yielding a value of 1,246,898 square µm ($A = 4\pi r^2$). We then modeled regions representing each half of the heart as square tiles with an equivalent surface area (height x width = 623,449 µm$^2$) and a depth of 97.5 µm, corresponding to the estimated distance from the epicardium to the endocardium. Groups of clonally related labeled cells were modeled as spheres with a radius of 54 µm, based on the average dimensions of the e13.5 clones. We then defined a discrimination cutoff of 50 µm (based on a conservative analysis of the maximum intra-clone distance between any labeled cell and its nearest neighbor within the same clone) as the minimum distance required between any cells from two different groups of clones such that those clones were considered to be distinct entities. Using these empirically determined parameters, we then performed simulations to estimate the underlying clone generation rate (and its corresponding clonality rate) that best fit the observed data as follows. First, we tested a range of suitable values (from 0.5 to 10.0) for the clone generation rate (m), which specifies the average number of clones that are stochastically generated in each simulated half heart. For any given value of the generation rate, we randomly simulated 10,000 half hearts, with the number of clones (X) assigned to each heart randomly determined according to the Poisson distribution ($\text{Prob}(X = k) = e^{-m} m^k / k!$). The centroids of each clone were then randomly assigned with uniform probability to locations within the volume of the simulated region, and any clones that were located closer than the allowed discrimination distance of each other were merged together to form a single 'cluster'. Then, the mean number of observed 'clusters' (clusters may either consist of a single clone or a set of neighboring merged clones) was calculated as a function of the clone generation rate. The underlying clone generation rate that best matched the experimental data with an average of 3.3 clusters per region was then chosen as the estimated true generation rate, and its associated clonality rate, defined as the fraction of observed clusters that are comprised of individual clones, was computed from the results of 10,000 simulations. The calculated clonality rates represent the likelihood that any individual cluster of labeled cells is indeed clonal, based on the results of this simulation analysis. Finally, we repeated the entire modeling process with different settings for the discrimination distance, in order to assess the robustness of the estimated clonality rate to different values for this parameter (*Figure 3—figure supplement 1*).

## Quantification of caSMC and pericyte numbers in adult hearts

Adult hearts (n = 3) were dissected and fixed in 4% paraformaldehyde (PFA) for one hour at 4°C, washed in PBS, and cryoprotected in 30% sucrose for 30 min. Hearts were then oriented apex-down within a mold of Tissue-Tek® O.C.T.™ compound before snap freezing and cryosectioning (20 µm). Sections were immunostained for SM-MHC and PDGFRβ as previously described, placed in Vectashield® with DAPI (Vector Labs, #H-1200), and frozen at -20°C overnight. A section from each heart that contained left ventricle, right ventricle, and septum tissue was chosen and 3 different fields of view (FOV) for each region of each heart were collected through 40X oil immersion microscopy. The total number of DAPI-labeled, PDGFRβ$^+$ cells signified pericytes in each FOV, while the total number of DAPI-labeled, SM-MHC$^+$ cells denoted smooth muscle cells.

### Lineage tracing

Both NG2-CreER and Notch3-CreER males were crossed to Rosa-$^{tdTomato}$ reporter mice. Cre activity was activated by tamoxifen that was dissolved in corn oil and delivered to pregnant dames by intraperitoneal injection at either e10.5 or 11.5 (4 mg) with identical results. Dames impregnated by NG2-CreER and Notch3-CreER males were sacrificed at e15.5. Embryonic hearts and kidneys were stained with SM-MHC, PDGFRβ and VE-cadherin as described above before each cell type was assessed for lineage labeling. NG2-CreER traced smooth muscle was observed in 10 hearts/kidneys from three

different litters. Notch3-CreER traced coronary smooth muscle was observed in 11 hearts and kidneys from three litters. The percentage of lineage labeled pericytes and caSMCs as shown in *Figure 4D* was quantified using Zeiss Zen imaging software to count individually labeled pericytes and ImageJ to measure the percentage of the linage label overlapping with SM-MHC immunoreactivity.

## PDGFRβ mutant analysis

A total of 12 mutant hearts from 5 litters were analyzed all of which showed the same phenotype. For *Figure 5C*, the number of Notch3[+] cells was recorded from a total of 7 wild type and 5 mutant hearts from 4 embryo litters from multiple 15,000 µm$^2$ fields of view: remodeling zone (Wt: n = 19 fields; -/-: n = 11 fields) and mature arteries (Wt: n = 11 fields; -/-: n = 11 fields).

## Notch3 mutant analysis

Embryos were collected from Notch3 heterozygous crosses, and the hearts immunostained for VEGFR2, SM-MHC, and PDGFRβ as described above. A total of 20 mutants hearts from 10 litters from time points e15.5 and 17.5 were analyzed, all of which showed the same phenotype. For quantification in *Figure 6F*, 9 mutant and 10 wild type hearts from 6 litters were analyzed (heterozygous hearts were not included in this analysis). Confocal z-stacks (with 14 µm intervals between z-planes) through the right lateral side of each heart were collected and imported into ImageJ (NIH). The 'segmented line' tool was used to measure the total length of the right coronary artery (and its auxiliary branches) covered either partially or fully by SM-MHC[+] cells. Discontinuous lengths of smooth muscle coverage were summed for each heart. Similar results were obtained when assessing SMα expression (n = 3 mutant hearts from three different litters).

To assess cell proliferation, mitotically active cells were measured through the incorporation of 5-ethynyl-2'-deoxyuridine (EdU). Pregnant females from Notch3 heterozygous crosses received a single intraperitoneal injection of 400 ug of Edu dissolved in 200 u L of dimethyl sulfoxide (DMSO). Three hours later, embryonic hearts were dissected, fixed, and immunostained for VEGFR2 and PDGFRβ as described above. EdU incorporation into DNA was detected through the Click-iT[®] EdU Alexa Fluor[®] 555 Imaging Kit performed at room temperature and using the protocol recommended by the manufacturer (Invitrogen, #C10338). A total of 4 wild type, 11 heterozygous, and 10 mutant hearts from 5 litters at time point e14.5 were analyzed. Confocal z-stacks at 20X magnification (with 3 um intervals between z-planes) through the right lateral side of each heart were collected and imported into Zeiss Zen imaging software. A 150 um x 150 um square was drawn at both the vascular plexus and at a coronary artery remodeling zone. The percentage of EdU-positive pericytes was determined by quantifying the number of cells in each region that were PDGFRβ[+] and EdU[+]-compared to only PDGFRβ[+].

## Isl1 mutant heart analysis

Embryos were collected at e13.5 from crosses between Isl1 heterozygous and wild type mice. The hearts were immunostained for Jagged-1, SM-MHC, and VE-Cadherin as described above. A total of 27 heterozygous and 26 wild type hearts from 5 litters were analyzed. Confocal z-stacks (with 14 µm intervals between z-planes) through the right lateral side of the each heart were collected and analyzed. For each heart, the relative fluorescence intensity of Jagged-1 was recorded as a range from absent or very low to very high expression.

## Statistics

Statistical analyses were performed using SigmaPlot version 12.0 (Systat Software Inc) or Prism (Graphpad), where appropriate normality and variation were calculated. Data are represented as mean ± standard deviation (sd). Mann-Whitney Rank Sum tests were performed as appropriate for two-group comparisons, and one-way ANOVA was performed for multiple-group (more than 2 groups) comparisons and post hoc analysis was used with a Holm-Sidak post hoc test. A p <0.05 was considered statistically significant. Samples sizes were chosen so that statistically significant values would be obtained.

## Acknowledgements

We thank Drs. Sylvia Evans (Tbx18-Cre and Isl1-Cre) and Silvia Fre (Notch3-CreER) for mouse strains, and Red-Horse lab members for their input and comments. KSV is supported by a CIRM graduate student research training grant (TG2-01159). AHJ was supported by a summer research grant administered by the Office of the Vice Provost for Undergraduate Education at Stanford University. KR is supported by the NIH (4R00HL10579303) and the Searle Scholars Program. ILW is supported by grants from the National Heart, Lung, and Blood Institute of the NIH under award number U01 HL099999; the California Institute for Regenerative Medicine (RT2-02060), and the Virginia and DK Ludwig Fund for Cancer Research, Dan P. Riordan was supported by a grant from the National Heart, Lung, and Blood Institute of the NIH (U01 HL099995).

## Additional information

### Funding

| Funder | Grant reference number | Author |
|---|---|---|
| National Institutes of Health | 4R00HL10579303 | Kristy Red-Horse |
| National Institutes of Health | U01 HL099999 | Irving Weissman |
| National Institutes of Health | U01 HL099995 | Daniel P Riordan |
| California Institute for Regenerative Medicine | RT2-02060 | Irving Weissman |
| Searle scholars program | | Kristy Red-Horse |

The funders had no role in study design, data collection and interpretation, or the decision to submit the work for publication.

### Author contributions

KSV, Conception and design, Acquisition of data, Analysis and interpretation of data, Drafting or revising the article, Contributed unpublished essential data or reagents; AHJ, Conception and design, Acquisition of data, Analysis and interpretation of data; HIC, AP, ASMcK, Acquisition of data, Analysis and interpretation of data; DPR, KRH, Conception and design, Acquisition of data, Analysis and interpretation of data, Drafting or revising the article; NK, JK, Drafting or revising the article, Contributed unpublished essential data or reagents; IW, Conception and design, Analysis and interpretation of data, Drafting or revising the article

### Ethics

Animal experimentation: All animal experiments were performed according to protocols approved by the Stanford University Institutional Animal Care and Use Committee (IACUC) under the protocol #26923 (Assurance #A3213-01). The laboratory animal care program at Stanford University is fully accredited by the Association for Assessment and Accreditation of Laboratory Animal Care International (AAALAC).

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
