## [Decision Letter]

Thank you for submitting your work entitled "Pericytes are progenitors for coronary artery smooth muscle" for peer review at *eLife*. Your submission has been favorably evaluated by Fiona Watt (Senior editor) and two of the three reviewers, one of whom is a member of our Board of Reviewing Editors. However, one of the reviewers raised substantial concerns about the lineage tracing specificity and the markers used to identify the two phenotypes. Although the other reviewers have only minor issues, relatively easy to address, it is important that in the revised version the concerns of the first reviewer are fully addressed.

We realise that this is a catch 22 situation, since "really specific" markers for pericytes do not exist, as correctly pointed out in the text; thus absence of a late marker (SMyHC) as only tool to distinguish caSMc from pericytes may be misleading. We strongly suggest that additional smooth muscle markers are used to distinguish immature caSMCs from pericytes. In general, the conclusions of the work should be toned down, outlining the complexity of the problem and how this work offers a possible answer, not the definitive one.

*Reviewer #1:* The authors aim to identify the intermediate progenitor cell type between the epicardium and coronary artery smooth muscle cells. Using clonal analysis of rare recombination events and genetic lineage tracking studies in mice, they propose that the intermediate progenitor is the pericyte. They also identify the role of PDGFRb and Notch3 for epicardium to pericyte and pericyte to caSMC transitions respectively. They then use lineage tracing and present a more limited dataset in which they propose that renal VSMC also originate from pericytes.

1) The major limitation in this paper is the use of non-specific markers for distinguishing pericyes from caSMCs. The authors use expression of NG2, PDGFRb and Notch3 to identify pericytes. However, each of these markers including NG2 can also be expressed by SMC (for example Murfee et al. Microcirculation; 2005: 151-160. doi: 10.1080/10739680590904955). Indeed a major challenge in the field of mural cells is the lack of genetic markers that unequivocally distinguish pericytes from smooth muscle cells as the authors own cited reference makes clear (Armulik et al. Dev Cell 2011). Given this constraint, it is difficult to be sure whether caSMC come from genuine pericytes or indeed whether the 'pericyte-like' cells actually represent an alternative SMC progenitor or immature SMCs. This is a major problem in the field and studies identifying unique pericyte markers are urgently required.

2) Given the lack of specificity of the markers employed, then the clonal analysis may simply suggest that caSMC and pericytes originate from a common progenitor. The later development of MYH11 expression could equally be due to the maturation of immature SMCs that were misinterpreted as being pericytes.

3) Another concern is that even if the NG2-CreER tracking was specific for pericytes (which this reviewer does not believe to be the case) then at least some of the MYH11+ cells do not have a NG2-derived origin (see Figure 4 – a substantial number of MYH11+ SMCs are not labelled green with the NG2-CreER lineage tracer). A quantitative assessment of the proportion of MYH11+ cells from the NG2 lineage would clarify what proportionof caSMCs have at least a common origin with pericytes and to what extent there is an entirely independent pathway for caSMC development.

4) MYH11 is a specific but not sensitive marker for developing SMCs since immature SMC may not express this relatively late onset marker. Thus it is not clear whether in the absence of Notch3, caSMC do not develop at all or whether maturation is impaired. This is an important difference and needs to be clarified by labelling with other markers such as ACTA2 and CNN1 and cell counting.

5) Finally, if the authors are correct about the progenitor role of pericytes, then is this a role they reprise in adults? Do pericytes contribute to new caSMC and vascular development in the collateral response to ischemia or during vessel regeneration following infarction in adult animals?

*Reviewer #2:* This is an excellent manuscript that finally solves a long lasting issue in developmental biology. Experiments are well planned and results compelling. I have only a few but relevant issues that should be addressed when revising the manuscript.

1) The size of figures is vanishingly small. This is obviously due to the need to compress a large number of images in a small space. Enlarging the PDF leads to a rapid loss of resolution. It would help to have a higher magnification of crucial images in supplementary figures (many of which are quite empty) that may help the reader to appreciate morphological details of co-localization that otherwise would require opening a different file.

2) Figure 2: The clonal nature of the labeled population is crucial for the interpretation of the data. The Authors show two clusters of cells (2A and 2B) and state that each "clone: is at least 100 micron apart from other labeled cells". It would be important to state how many clones are detected in the heart and provide some evidence, direct or indirect, that the probability of labeling two neighboring (but not necessarily sister) cells is indeed null or sufficiently small not to alter data interpretation.

3) Figure 3: If pericytes are progenitors of caSMCs, one would expect that with time the ratio between progenitor and differentiated cells would shift towards the latter. The fact that, even in adult heart, there are many more pericytes suggests that either pericytes largely outnumber SMCs or that caSMCs may also have other, different progenitors. Authors should clarify this issue by showing that the ratio in the clones is consistent with the ratio pericytes/caSMAc in normal adult heart (or not: equally interesting and easy to check).

4) NG2 is expressed in developing cardiomyocytes. Authors state that control experiments rule out the possible origin of caSMCs from caridiomyocytes (data not shown) but should at least state which Cre driver they used for the control experiments.

*Reviewer #3:* The main objective of the study is to ascertain the origin of smooth muscle cells in coronary arteries. The authors use a series of elegant cell tracing and clonal analysis to clarify the intermediary steps between epicardial invasion and smooth muscle cell differentiation. They conclude that coronary smooth muscle cells originate from cells that are PDGFRb/NG2/Notch3 positive and recognized as "pericytes" because of this expression profile. These cells associate with the endothelial plexus and further differentiate (express SMHHC) into vascular smooth muscle cells at sites of arterial remodeling and in response to flow forces. Overall the study is very well performed, extremely well documented / supported and clearly written. I only have some suggestions / conceptual concerns to express (as indicated below) that can be easily addressed by simple editorial changes in the Results/Discussion.

1) It is important to highlight that the terminology of these cells as "pericytes" is based on expression of NG2(?). Do these cells fully associate with capillaries (?) for some period prior to differentiation into VSMCs? And/or associate with capillaries in a sheet of basement membrane? Do they develop the typical phenotype of pericytes with long/branched processes? Or are these more like mesenchymal cells that can either give rise to pericytes or VSMC depending on flow?

2) Do the authors know whether adult pericytes retain the capacity to differentiate into VSMC? In other words, what is the origin of VSMC in collaterals induced post-ischemia?

3) The comments related to the contribution of flow are more "correlative" than mechanistic. It would be really nice to see if ligation/alteration of flow would affect the association or differentiation of these cells. However, these experiments were not performed and, therefore, the link with flow should be toned down.

---

## [Author Response]

*One of the reviewers raised substantial concerns about the lineage tracing specificity and the markers used to identify the two phenotypes. Although the other reviewers have only minor issues, relatively easy to address, it is important that in the revised version the concerns of the first reviewer are fully addressed. We realise that this is a catch 22 situation, since "really specific" markers for pericytes do not exist, as correctly pointed out in the text; thus absence of a late marker (SMyHC) as only tool to distinguish caSMc from pericytes may be misleading. We strongly suggest that additional smooth muscle markers are used to distinguish immature caSMCs from pericytes. In general, the conclusions of the work should be toned down, outlining the complexity of the problem and how this work offers a possible answer, not the definitive one.*

To summarize the general changes requested above:

1) Additional SMC markers (Smooth muscle actin and Calponin) were used to further characterize pericytes and smooth muscle cells (revised Figure 2, and K and Figure 2—figure supplement 2). Smooth muscle actin and Calponin were not expressed in what we had identified as pericytes in the first submission. These data add additional support that PDGFRβ+SM-MHC- cells located around microvessels are pericytes and not a separate population of immature smooth muscle progenitors. Interestingly, we found that SM-MHC (SMyHC) was not a late smooth muscle marker in the developing heart. It was the first of the three smooth muscle markers to be expressed in caSMCs being present in early remodeling arteries in addition to large, mature arteries, that latter of which also expressed Smooth muscle actin and Calponin.

2) Throughout the text, we added points and changed the wording to tone down conclusions.

Reviewer #1:

*1) The major limitation in this paper is the use of non-specific markers for distinguishing pericyes from caSMCs. The authors use expression of NG2, PDGFRb and Notch3 to identify pericytes. However, each of these markers including NG2 can also be expressed by SMC (for example Murfee et al. Microcirculation; 2005: 151-160. doi: 10.1080/10739680590904955). Indeed a major challenge in the field of mural cells is the lack of genetic markers that unequivocally distinguish pericytes from smooth muscle cells as the authors own cited reference makes clear (Armulik et al. Dev Cell 2011). Given this constraint, it is difficult to be sure whether caSMC come from genuine pericytes or indeed whether the 'pericyte-like' cells actually represent an alternative SMC progenitor or immature SMCs. This is a major problem in the field and studies identifying unique pericyte markers are urgently required.* The perivascular cells in our study were termed pericytes because we found no evidence that a subset of these were distinct pericyte-like caSMC progenitors. The point that our studies cannot distinguish between a pericyte as we have defined it (Notch3^+^PDGFRβ^+^NG2^+^SM-MHC^-^SMα–PDGFRα^-^, wrapping small blood vessels, and within the basement membrane) and a”pericyte-like” SMC progenitor with these exact same attributes is correct. However, by eight criteria (additional markers added in revised Figure 2, and K and Figure 2—figure supplement 2), which includes a morphology and tissue localization that is not shared with SMCs, they appear to be pericytes. Given our extensive marker analysis, we believe our data supports the model that genuine pericytes are themselves SMC progenitors. Even if future research with additional markers showed that there is a “pericyte-like” SMC progenitor different from, but intermixed with, traditional pericytes, we believe our study would remain an important scientific advance in that it is the first to identify the cellular pathway from the epicardium to coronary artery. Specifically, epicardial derived pericytes or “pericyte-like” cells travel along blood vessels to find coronary arteries, which is critical information when searching for localized signals guiding caSMC development. We have added new text to address this point, including discussing that we cannot distinguish a “pericyte-like” progenitor from a traditional pericyte is they both display the following profile: Notch3^+^PDGFRβ^+^NG2^+^SM-MHC^-^SMα–PDGFRα^-^, extended processes around small blood vessels, and exist within the basement membrane. Changes to the text have been made to the Introduction, Results, and Discussion.

*2) Given the lack of specificity of the markers employed, then the clonal analysis may simply suggest that caSMC and pericytes originate from a common progenitor. The later development of MYH11 expression could equally be due to the maturation of immature SMCs that were misinterpreted as being pericytes.*

We have used not only molecular markers, but morphological characteristics, i.e. wrapping small vessels and embedded within the basement membrane. By eight criteria the cells appear to be pericytes. Current studies in our laboratory are aimed at finding markers that distinguish cardiac pericytes and coronary smooth muscle. If further research allows us to distinguish a pericyte-like caSMC progenitor from pericytes, we will still have made an important discovery: that epicardial cells take a perivascular route to coronary arteries and that this progenitor looks very similar to pericytes by multiple criteria. Since we have found no evidence thus far that some of the perivascular cells are distinct pericyte-like caSMC progenitors, we believe it makes sense to identify the cells as pericytes.

*3) Another concern is that even if the NG2-CreER tracking was specific for pericytes (which this reviewer does not believe to be the case)…*

To be as thorough as possible with the tools available, we used two lineage tracing reagents for this experiment (NG2-CreER and Notch3-CreER) and extensively analyzed NG2 and Notch3 expression in perivascular cells (see fourth paragraph of subsection “Clonal analysis of epicardial-derived cells “and revised Figure 2). We concluded that these two reagents would provide good evidence that pericytes/pericyte-like cells give rise to caSMCs.

*…then at least some of the MYH11+ cells do not have a NG2-derived origin (see*
Figure 4*– a substantial number of MYH11+ SMCs are not labelled green with the NG2-CreER lineage tracer). A quantitative assessment of the proportion of MYH11+ cells from the NG2 lineage would clarify what proportionof caSMCs have at least a common origin with pericytes and to what extent there is an entirely independent pathway for caSMC development.*

In the revised manuscript, lineage labeling was quantified in both pericytes and caSMCs by NG2-CreER and Notch3-CreER (revised Figure 4). The percentage of traced cells was comparable between both cell types using both Cre lines. Since NG2 and Notch3 were expressed in all developing cardiac pericytes and, as shown in other systems, reporter gene recombination efficiency is usually not complete with CreERs, we concluded that the incomplete labeling of caSMCs is likely due to low recombination efficiency. An alternative, and not mutually exclusive, explanation is that an additional population contributes to caSMCs. Furthermore, the existence of an independent source of caSMCs does not diminish the importance of the pathway studied here. This data was added to revised Figure 4 and the Results section.

*4) MYH11 is a specific but not sensitive marker for developing SMCs since immature SMC may not express this relatively late onset marker. Thus it is not clear whether in the absence of Notch3, caSMC do not develop at all or whether maturation is impaired. This is an important difference and needs to be clarified by labelling with other markers such as ACTA2 and CNN1 and cell counting.*

We have addressed this concern with a more extensive analysis of Notch3 mutants (revised Figure 7). We added PDGFRβ and SMα staining as well as cell counting and proliferation rates. To summarize, mural cell coverage, cell numbers, and proliferation rates were similar between WT, HET, and KO at microvessels and larger arteries. The distinction is in SM-MHC and SMα expression, which were both decreased in mutant hearts. We concluded that Notch3 is responsible for inducing the expression of contractile proteins, and therefore caSMC maturation. Changes in text have been made to the Results.

*5) Finally, if the authors are correct about the progenitor role of pericytes, then is this a role they reprise in adults? Do pericytes contribute to new caSMC and vascular development in the collateral response to ischemia or during vessel regeneration following infarction in adult animals?*

We are very interested in this question and experiments to address it are underway. However, these studies are genetically complicated and will take some time to complete. They are planned for another publication.

Reviewer #2:

*This is an excellent manuscript that finally solves a long lasting issue in developmental biology. Experiments are well planned and results compelling. I have only a few but relevant issues that should be addressed when revising the manuscript. 1) The size of figures is vanishingly small. This is obviously due to the need to compress a large number of images in a small space. Enlarging the PDF leads to a rapid loss of resolution. It would help to have a higher magnification of crucial images in supplementary figures (many of which are quite empty) that may help the reader to appreciate morphological details of co-localization that otherwise would require opening a different file.*

Full resolution images were uploaded, which are clearer when increased in size. Higher magnifications were included throughout the figures, specifically revised Figure 2 – right panels, 2F, Figure 1—figure supplement 1, and Figure 2—figure supplement 4.

*2) Figure 2: The clonal nature of the labeled population is crucial for the interpretation of the data. The Authors show two clusters of cells (2A and 2B) and state that each "clone: is at least 100 micron apart from other labeled cells". It would be important to state how many clones are detected in the heart and provide some evidence, direct or indirect, that the probability of labeling two neighboring (but not necessarily sister) cells is indeed null or sufficiently small not to alter data interpretation.*

We collaborated with a Bioinformatics colleague, Dr. Daniel Riordan, who performed mathematical modeling based on our experimentally observed cluster frequency. His simulations showed that greater than 80% of the clusters are likely clonal. Given the number of clusters analyzed, this rate of clonality allowed us to uphold our initial conclusions regarding lineage relationships. Data was included in new Figure 3 and Figure 3—figure supplement 1. Associated text is in the Results section.

*3) Figure 3: If pericytes are progenitors of caSMCs, one would expect that with time the ratio between progenitor and differentiated cells would shift towards the latter. The fact that, even in adult heart, there are many more pericytes suggests that either pericytes largely outnumber SMCs or that caSMCs may also have other, different progenitors. Authors should clarify this issue by showing that the ratio in the clones is consistent with the ratio pericytes/caSMAc in normal adult heart (or not: equally interesting and easy to check).*

We quantified the ratio of pericytes to caSMCs in the adult heart and found the numbers to be not significantly different from the ratios of these same cells within clones. This data does not exclude the possibility of other sources, but, in addition to epicardial lineage tracing, provides evidence that the epicardial/pericyte/caSMC pathway is a major contributor to smooth muscle. These data are shown in Figure 3 and in the text, in the Results section.

*4) NG2 is expressed in developing cardiomyocytes. Authors state that control experiments rule out the possible origin of caSMCs from caridiomyocytes (data not shown) but should at least state which Cre driver they used for the control experiments.*

The use of the Myh6-Cre driver was added to the text, Results, subsection “Lineage tracing pericytes in the developing heart”.

Reviewer #3:

*[…] I only have some suggestions / conceptual concerns to express (as indicated below) that can be easily addressed by simple editorial changes in the Results/Discussion. 1) It is important to highlight that the terminology of these cells as "pericytes" is based on expression of NG2(?). Do these cells fully associate with capillaries (?) for some period prior to differentiation into VSMCs? And/or associate with capillaries in a sheet of basement membrane? Do they develop the typical phenotype of pericytes with long/branched processes? Or are these more like mesenchymal cells that can either give rise to pericytes or VSMC depending on flow?*

Our terminology is based on the expression of Notch3, PDGFRβ, NG2 and the absence of SM-MHC, SMα and PDGFRα. Furthermore, we defined pericytes as closely wrapping small blood vessels, and being embedded within the basement membrane. These points were highlighted throughout the text. Pericytes associate with capillaries as soon as they differentiate from the epicardium at the cell surface and have long processes that travel along the vessels. They also do not form a continuous covering of the endothelium as SMCs do. When differentiating into vSMCs, pericytes acquire SMC contractile proteins (SM-MHC and SMα) and change their morphology (no longer exhibit long skinny processes and orient radially to form a continuous covering around large vessels).

*2) Do the authors know whether adult pericytes retain the capacity to differentiate into VSMC? In other words, what is the origin of VSMC in collaterals induced post-ischemia?*

We are very interested in this question and experiments to address it are underway. However, these studies are genetically complicated and will take some time to complete. They are planned for another publication.

*3) The comments related to the contribution of flow are more "correlative" than mechanistic. It would be really nice to see if ligation/alteration of flow would affect the association or differentiation of these cells. However, these experiments were not performed and, therefore, the link with flow should be toned down.*

The below alterations were made to the text to address this concern:

Abstract: “Blood flow-induced” was deleted.

Introduction: “We show that pericytes lining the coronary vascular plexus respond to (“blood flow induced” deleted here) Notch3 signaling at arterial remodeling zones to become caSMCs during embryonic development. […] Thus, blood flow initiation is correlated with molecular changes that mark the site of arterial development in the coronary vascular plexus and may stimulate vessel remodeling and pericyte differentiation into smooth muscle.”